# Low-Temperature Plasma-Activated Medium Inhibits the Migration of Non-Small Cell Lung Cancer Cells via the Wnt/*β*-Catenin Pathway

**DOI:** 10.3390/biom13071073

**Published:** 2023-07-04

**Authors:** Yan Zhang, Zhuna Yan, Hui Wu, Xiao Yang, Ke Yang, Wencheng Song

**Affiliations:** 1School of Medicine, Anhui University of Science and Technology, Huainan 232001, China; zhangdapangha@163.com (Y.Z.); yanzhuna0663@163.com (Z.Y.); 2Anhui Institute of Optics and Fine Mechanics, Institute of Health & Medical Technology, Hefei Institutes of Physical Science, Chinese Academy of Sciences, Hefei 230031, China; wh18326114800@126.com (H.W.);; 3Hefei Cancer Hospital, Chinese Academy of Sciences, Hefei 230031, China; 4Collaborative Innovation Center of Radiation Medicine of Jiangsu Higher Education Institutions and School for Radiological and Interdisciplinary Sciences, Soochow University, Suzhou 215123, China

**Keywords:** plasma-activated medium, NSCLC, migration, Wnt/*β*-catenin, RONS

## Abstract

This study explored the molecular mechanism of the plasma activation medium (PAM) inhibiting the migration ability of NSCLC (non-small cell lung cancer) cells. The effect of PAM incubation on the cell viability of NSCLC was detected through a cell viability experiment. Transwell cells and microfluidic chips were used to investigate the effects of PAM on the migration capacity of NSCLC cells, and the latter was used for the first time to observe the changes in the migration capacity of cancer cells treated with PAM. Moreover, the molecular mechanisms of PAM affecting the migration ability of NSCLC cells were investigated through intracellular and extracellular ROS detection, mitochondrial membrane potential, and Western blot experiments. The results showed that after long-term treatment with PAM, the high level of ROS produced by PAM reduced the level of the mitochondrial membrane potential of cells and blocked the cell division cycle in the G2/M phase. At the same time, the EMT process was reversed by inhibiting the Wnt/*β*-catenin signaling pathway. These results suggested that the high ROS levels generated by the PAM treatment reversed the EMT process by inhibiting the WNT/*β*-catenin pathway in NSCLC cells and thus inhibited the migration of NSCLC cells. Therefore, these results provide good theoretical support for the clinical treatment of NSCLC with PAM.

## 1. Introduction

Despite the rapid development of medicine today, the threat of cancer to human health continued to increase. According to research papers published in the International Journal of Cancer, there was an estimated 18.1 million new cancer cases and 9.6 million cancer deaths worldwide in 2018 [1]. It has been reported that lung cancer was the cancer with the highest incidence and mortality rates in China, with the number of new cases and deaths increasing from 698,000 and 554,000 in 2015 to 823,000 and 723,000 in 2020, respectively [2]. The high mortality rate of lung cancer was closely associated with its early metastatic characteristics, which is the ability of cancer cells to leave the primary focus and infiltrate other parts of the body [3,4]. Lung cancer could be divided into small cell lung cancer (SCLC) and non-small cell lung cancer (NSCLC), of which NSCLC accounted for about 85% of the total number of lung cancers [5]. The traditional treatments of NSCLC mainly included surgical resection, radiotherapy [6], chemotherapy [7], and targeted therapy [8]. Radiotherapy was an effective treatment for NSCLC and could be used in all stages of curative or palliative treatment [9]. Currently, platinum chemotherapy drugs were the most important chemotherapy means for advanced NSCLC patients and had a significant effect on prolonging the survival time of patients [10]. However, the inhibition effect of traditional treatment on cancer cell migration was not as expected. The existence of drug resistance and unbearable side effects forced us to find a new and effective method to inhibit NSCLC cell metastasis [11].

Plasma was the fourth state of existence of a substance which consisted mainly of positively charged ions, electrons, and neutral particles [12]. The plasma was divided according to temperature into high-temperature plasma and low-temperature plasma (LTP) [13]. LTP has been widely used in the biomedical field in recent years due to its advantages such as low temperature and lack of harm caused to the normal tissues of the human body. It was mainly applied to such areas as wound healing [14], oral treatment [15], microbial inactivation, and cancer treatment [16,17]. At present, low-temperature plasma medicine is widely used in dozens of cancers, such as skin cancer [18], melanoma [19], and colon cancer [20]. There are two main ways for LTP to treat cells. One is to directly treat the cells with LTP, and the other is that LTP is first used to treat a cell culture medium, and then the cells are incubated with the plasma-activated medium (PAM) [21]. Studies confirmed that the active substances in PAM could enter the cells to have an effect on their biology [22], such as destroying cell DNA and promoting cell apoptosis [23]. The active substances in PAM mainly included reactive oxygen species (ROS) and reactive nitrogen species (RNS). Hydrogen peroxide (H_2_O_2_) and nitrite (NO_2_^−^) were the main long-acting substances of RONS, and their intracellular interaction was an important reason for the selective apoptosis of tumor cells [24]. Recently, PAM has been widely studied for antitumor research, proving that its production of a large number of ROS was the main factor in cancer cell apoptosis [22,25,26]. In addition, PAM has been found to inhibit the migration of melanoma cells [27], but whether it inhibits the migration of NSCLC and its mechanisms has not been explored.

Cancer metastasis is a complex process. When cancer cells lose polarity, intercellular adhesion decreases and tight junctions are further lost, thus obtaining the mesenchymal phenotype. This process is called epithelial–mesenchymal transition (EMT), which is characterized by changes in the levels of three prominent biomarkers (E-Cadherin, Vimentin, and N-Cadherin). E-Cadherin expression is down-regulated, while the up-regulation of Vimentin and N-Cadherin leads to the occurrence of EMT [28,29]. The EMT process of cells is regulated by a variety of signaling pathways, among which the Wnt/*β*-catenin signaling pathway is the most closely related [30]. Moreover, the Wnt/*β*-catenin pathway also plays an important role in regulating cell proliferation, as well as cell migration and invasion [31,32,33]. Multiple studies have shown that the proliferation and metastasis of cancer cells such as lung cancer and bladder cancer are effectively inhibited by inhibiting the Wnt/*β*-catenin signaling pathway [34,35,36,37,38]. Furthermore, Liu et al. also proved that chromatin remodeling ATPase SMARCAD1 promoted the EMT process by activating the Wnt/*β*-catenin signaling pathway, thereby promoting the metastasis of pancreatic cancer cells [39]. However, whether PAM could inhibit the migration of NSCLC cells by down-regulating the Wnt/*β*-catenin pathway in cancer cells and inhibiting the EMT process has not been explored, and our research was dedicated to this. 

In this study, the effects of PAM on the cell migration of NSCLC were detected at the in vitro level using microfluidic chips and Transwell cells. The expression levels of EMT and Wnt/*β*-catenin-signaling-pathway-related proteins were detected via Western blot experiments to explore the internal mechanism of PAM affecting cell migration.

## 2. Materials and Methods

### 2.1. Cell Culture

NSCLC cells H460, H1299, A549, PC-9, and H1975 were purchased from the ATCC cell bank. The NSCLC cells were cultured in RPMI medium 1640 Basic (GIBCO, Carlsbad, CA, USA) containing 10% fetal bovine serum (FBS, Longsera, Shanghai, China) and 1% penicillin/streptomycin (NCM BioTECH, Suzhou, China). Cells were cultured in a 37 °C cell incubator (Thermo Fisher Scientific, Waltham, MA, USA) with 5% CO_2_. When the cells reached approximately 80% plate bottom coverage, they were incubated in LTP-treated medium for 0 s, 10 s, 15 s, 20 s, 25 s, and 30 s.

### 2.2. LTP Device

The DBD dielectric barrier discharge device used in this study was described in Zhou et al. [40]. The effective voltage of this device was 3.78 kV, and the frequency was 25 kHz. Pure helium (He, 99.999%) was used as the discharge gas at a flow rate of 1 L/min. The device had to be kept closed when the medium was treated with the device. Before treatment, helium gas was introduced for 90 s, and then the culture medium was subjected to LTP treatment for different amounts of time.

### 2.3. Cell Viability

3-(4, 5-Dimethyl-2-thiazolyl)-2, 5-diphenyltetrazole bromide (MTT, Sigma-Aldrich, St. Louis, MO, USA) was used as a cell viability assay. There were five NSCLC cell lines, including H460, PC-9, A549, H1299, and H1975. When the cells grew to be suitable for passage, the adherent cells were digested with trypsin and added to cell culture solution to make a cell suspension, and then transferred to each Petri dish evenly. When the cells in each dish grew to about 80% of the bottom of the dish, they were incubated with PAM for 24 h. After the PAM was discarded, 1 mL of pre-prepared MTT working solution was added into each dish for continuous incubation for 4 h. The plates were removed from the MTT working solution and an equal volume of dimethyl sulfoxide (DMSO, Biotechnology, Shanghai, China) was added to each plate to dissolve the crystals in the cells. After sufficient shaking, 150 μL of dissolved liquid was added to each well in a 96-well plate, and a microplate reader was used to determine the absorbance at 492 nm for each well. Cell viability was calculated according to a standard curve. 

### 2.4. Wound Healing Assay

H460 and PC-9 cells were incubated in 10% fetal bovine serum until the bottom of the plate fused. The original culture medium in the Petri dish was discarded, and the monolayer on the cell fusion surface was scraped off with a 20 μL white pipette head. The scraped cells and cell fragments were washed with PBS three times, and the cells were incubated in PAM without FBS. Cell migration of PAM after incubation for 4 h, 12 h, 24 h, and 48 h was observed and recorded using an inverted microscope (Guangzhou, China), and the scratch surface area after the different incubation times was quantified using different gray values.

### 2.5. Transwell Assay

In addition to the scratch experiment, this study also used the Transwell tool to further explore cell migration and analyze the influence of PAM on cell migration. The experiment was carried out in an 8 μm 24-hole Transwell plate (Costar, Washington, DC, USA). H460 and PC-9 cells were incubated in PAM for 4 h, digested by trypsin, and added to the FBS-free medium to prepare a cell suspension. A cell count was carried out to make the cell concentration of the cell suspension about 1 × 10^5^/mL. In total, 500 μL 10% FBS medium was added into the outer chamber of the Transwell plate, and 200 μL cell suspension was added into the inner chamber. The cells were fixed and stained after incubation for 24 h. The inner chamber was wiped and observed under an inverted microscope. Five fields of view were randomly selected.

In addition, to further explore whether the inhibition of high-dose PAM treatment (30 s) on the migration of H460 cells and PC-9 cells was achieved by down-regulating the Wnt/*β*-catenin signaling pathway, we added IWP-2, an inhibitor of the Wnt/*β*-catenin pathway (Beyotime, Shanghai, China), in a complementary migration experiment. The experiment was divided into four groups, namely, the control group, the PAM (30 s) treatment group, the IWP-2 treatment group, and the PAM (30 s) + IWP-2 treatment group. The working concentration in the IWP-2 treatment group was 12.5 μL, and the specific steps in the migration experiment were the same as those above.

### 2.6. Microfluidic Chip Preparation

The microfluidic chip was designed using AutoCAD (Autodesk, San Rafael, CA, USA). A single-cell migration testing unit included a cell loading unit, a chemotactic migration unit, and a cell separation unit. The fabrication steps of microfluidic chips were described in detail in a previous study [41]. The master mold contained two layers. The first layer was designed to be 3 μm thick for generating the cell arrangement areas. The second layer was designed to be 70 μm thick for developing the main channels. After the master mold fabrication was completed, polydimethylsiloxane (PDMS) was injected and incubated at 70 °C for 2 h. The PDMS replica was then cut out. The PDMS with punched-out holes was bonded to a glass slide using the plasma cleaner (Harrick, America). Before the experiments, the microfluidic chip was filled with the cell adhesion agent (rat-tail type Ⅰ collagen, Meilun Biology, Dalian, China) and incubated at 37 °C for 60 min. The cell adhesion agent was removed from the chip, where it was re-filled with the cell culture medium (0.4% bovine serum albumin diluted with RPMI-1640, glucose-free) and incubated at 37 °C for another 60 min.

Six independently controlled cell migration test units were configured on a single chip, allowing parallel testing for different conditions (Figure 1A–C). The simulation software Comsol Multiphysics 5.5 was used to simulate the generation of chemical concentration gradients in the main channels of single cell migration [41]. The simulation parameters in the software were as follows: initial concentrations of chemokine inlet and cell culture medium inlet were set to be C1 = 1 mol/(L m^3^) and C2 = 0 mol/(L m^3^), respectively; liquid density was 10^3^ kg/m; dynamic viscosity was 10^–3^ Pa s; and diffusion coefficient D was 7.5 × e^–11^ m^2^/s. The liquid height difference at the inlet and the outlet of the microfluidic concentration gradient chip pipeline could form a pressure difference inside the pipeline to promote fluid flow. The height of the chip body was set to 5 mm so that the pressure at the chemokine injection port and the cell culture medium injection port could be set to 50 Pa at most, and the pressure at the output port was set to 0 Pa assuming an infinite diameter and no liquid at the outlet. Identical and stable chemical gradients could be rapidly generated in the six test channels without requiring external pumps for up to 48 h, which was sufficient for the cancer cell migration experiments. For the cancer cell experiments, the chemoattractant solution and medium were topped up intermittently to the inlet wells to maintain the stable gradient for a longer period of time. The cell docking structure aligned cells next to the barrier channel before the chemical gradient exposure, which simplified cell migration assay operation and analysis and improved the accuracy. It could simply involve cell seeding and solution loading followed by incubation and final end-point imaging analysis, while still offering the option of real-time monitoring and cell tracking analysis via time-lapse microscopy.

### 2.7. Microfluidic Cell Migration Experiments

PC-9 cells were divided into a control group and a 30-spam treatment group. Then, the cells were loaded into six parallel test units of the microfluidic chip from the cell inlet. PC-9 cells in the control group were injected into channels I, II, and III, and PC-9 cells in the 30 s PAM treatment group were injected into channels IV, V, and VI. An equal volume of chemical attraction solution (IL-8, SIGMA-Aldrich, USA) at a concentration of 200 ng/mL and cell culture medium (1% 1640) were injected into the cell chemotaxis unit and the cell culture medium unit, respectively. The microfluidic device was then placed onto the imaging stage of an inverted microscope.

For the cancer cell migration experiment, the images of cells in the device were captured every 6 h for a total of 24 h and the device was kept in a 37 °C incubator containing 5% CO_2_ between the imaging time points. The microscope was equipped with an environmental control chamber to maintain the microscope stage at 37 °C. It is worth noting that the six migration units were captured by connecting a smartphone camera to the microscope eyepiece.

The calculation of cell migration distance (cell initial position) along the gradient from the docking region was mentioned in a previous document [42]; 1 × 10^5^ cells were analyzed for each experiment. 

### 2.8. Extracellular Active Substance Detection

The extracellular active substances in PAM were mainly reactive oxygen species (ROS) and reactive nitrogen species (RNS). Therefore, extracellular ROS and RNS were detected by using a H_2_O_2_ detection kit (Beyotime, Shanghai, China) and NO detection kit (Beyotime, Shanghai, China), respectively. According to the kit instructions, 50 μL of PAM was added to each well in a 96-well plate for incubation for different times, followed by the detection of ROS and RNS concentrations in the medium with H_2_O_2_ and NO detection reagents. The concentrations of H_2_O_2_ and NO in the culture medium were detected again after the cells were incubated with PAM for 4 h. The absorbance was determined at 540 nm and the concentration was calculated from the standard curve.

### 2.9. Intracellular Active Substance Detection

In this experiment, ROS in the cells incubated with PAM was detected using a reactive oxygen species detection kit (KeyGEN BioTECH, Nanjing, China). H460 and PC-9 cells were incubated with PAM at 37 °C for 4 h and then with the addition of a DCFH-DA fluorescent probe for 30 min. After incubation, the cells were washed three times, and observed and recorded using a fluorescence microscope (Olympus, Tokyo, Japan). 

In addition, to explore whether the inhibition of NSCLC cell migration via high-dose PAM is related to the high level of ROS produced, a group of ROS verification experiments were performed on PC-9 cells. N-acetyl-l-cysteine (NAC, Beyotime, Shanghai, China) was added 1 h before the LTP-treated cell culture medium at a working dose of 10 mM, and the time gradients of LTP-treated PC-9 cells were 0 s, 10 s, 20 s, and 30 s. To observe the effect of NAC pretreatment on cell migration ability, Transwell chamber and scratch experiments were used for verification. The specific experimental steps were the same as those mentioned above. 

### 2.10. Detection of Cell Mitochondrial Membrane Potential

The mitochondrial membrane potential detection kit (JC-1, Solarbio, Beijing, China) was used to detect the intracellular mitochondrial membrane potential after cell PAM treatment. After the lung cancer cells H460 and PC-9 were incubated with PAM for 4 h, the old medium was discarded and washed with PBS. A mixture of 1 mL of medium and 1 mL of JC-1 working solution was then added to each dish and incubation continued for 20 min at 37 °C. After incubation, the samples were washed with JC-1 staining buffer, followed by the addition of the medium to each dish and the green and red fluorescence were observed and recorded using fluorescence microscopy (Olympus, Tokyo, Japan).

### 2.11. Cell Cycle Experiment

The lung cancer cells treated with PAM were subjected to cell cycle testing according to the cell cycle testing kit (KGI Biology, Nanjing, China). After PC-9 cells were incubated with PAM for 24 h, they were digested with trypsin to prepare the cell suspension, which was collected into each tube for centrifugation. After washing twice with PBS and centrifugation, the supernatant was added with 1 mL of 70% ethanol and stored at 4 °C overnight. Each tube was added with 500 μL staining solution (RNase: propidium iodide = 1:9) in a dark configuration, stained for 30 min, and cell cycle was detected using flow cytometry (BD Biosciences, Franklin Lakes, NJ, USA).

### 2.12. Western Blot

PC-9 cells were collected after 24 h incubation with PAM. PC-9 cells were lysed according to the kit description, and the BCA protein detection kit (Beyotime, Shanghai, China) was used to quantitate the extracted protein. Electrophoresis was performed and Western blot analysis was performed using a gel prepared with the SDS-PAGE gel kit (Beyotime, Shanghai, China). After 10 μL protein was added to each electrophoresis tank, the protein was transferred to nitrocellulose membrane (Beyotime, Shanghai, China) after electrophoresis. After 2 h of blocking with 5% defatted milk, rabbit anti-Wnt3A antibodies (ABCONE, Wuhan, China, 1:1000), rabbit anti-*β*-catenin antibodies (ABCONE, Wuhan, China, 1:1000), rabbit anti-VEGF antibodies (AB Clone, Wuhan, China, 1:1000), and rabbit anti-VEGFR 2 antibodies (AB Clone, Wuhan, China, 1:1000) were detected. Rabbit anti-N-Cadherin antibodies (AB Clone, Wuhan, China, 1:1000), rabbit anti-E-Cadherin antibodies (AB Clone, Wuhan, China, 1:1000), rabbit anti-Vimentin antibodies (AB Clone, Wuhan, China, 1:1000), rabbit anti-c-Jun antibodies (Cell Signaling, Danvers, MA, USA, 1: 1000), and rabbit anti-Cyclin D1 antibodies (Cell Signaling, Danvers, MA, USA, 1: 1000) were incubated overnight at 4 °C, respectively. Post-horseradish peroxidase (HRP, ABCONE, Wuhan, China, 1; 10,000) was used as a second antibody for 40 min. Finally, luminescence imaging was performed using a chemiluminescent kit (Scientific Seal Le Scientific, Thermo Fisher Scientific, Waltham, MA, USA) and a chemiluminescent gel imaging system (Tanon, Shanghai, China). The blot was quantified using a gray value detection method.

### 2.13. Data Analysis

The data are reported as the average standard deviation of three independent experiments. A *t*-test was used for comparison between the two groups, and Bonferroni’s corrected one-way analysis of variance (ANOVA) and two-way analysis of variance (two-way ANOVA) were used for comparison between the two groups. The statistically significance difference was based on * *p* < 0.05, ** *p* < 0.01, *** *p* < 0.001.

## 3. Results 

### 3.1. Effect of PAM Treatment on the Viability of NSCLC Cells

As shown in Figure 2, the cell viability decreased in a dose-dependent manner with the increase in LTP treatment. In the 30 s PAM-treated group, H460 cell viability was 78%, PC-9 cell viability was 76%, while A549 cell viability was 94%, H1299 cell viability was 46%, and H1975 cell viability was 85%. When A549 cells were treated for an extended period of time up to 50 s, the cell viability was 86%. The results showed that different NSCLC cells had different sensitivities to PAM.

According to previous studies, when the cell viability was less than 75%, the unavoidable reduction in cell viability induced by PAM will decrease the cell migration and invasion ability [43]. To avoid this problem, the LTP exposure time in this experiment was selected to be 30 s or less. In addition, according to the results, for the subsequent experiments we selected H460 and PC-9 lung cancer cells to conduct, and determined that the time for cell treatment with LTP in the subsequent experiments was 0 s, 10 s, 15 s, 20 s, 25 s, and 30 s.

### 3.2. Effect of PAM Treatment on the Cell Migration of NSCLC Cells

The cell scratch test and the Transwell tool were used to examine the effect of PAM incubation on cell migration in the H460 and PC-9 cells. As shown in Figure 3A–D, after continuous observation for 48 h under the light microscope, the scratch areas of cells in the short-term PAM (10 s) treatment group were significantly reduced, while those in the long-term PAM treatment group (30 s) showed no significant difference as compared with those in the control group. The same results were observed in Transwell chamber (Figure 3E–H). Compared with the control group, the number of migrated cells in the short-term PAM treatment group increased to a certain extent, while the number of migrated cells in the long-term PAM treatment group was significantly less, and the difference was statistically significant. This phenomenon indicated that, with the further increase in treatment time, PAM inhibited the migration of NSCLC cells, and the inhibition of NCSLC cell migration was most significant in the long-term PAM treatment group.

The microfluidic chips could simulate the physiological environment in vivo and more truly reflect the migration status of NSCLC cells during the incubation process of PAM. PC-9 cells in the control group and the 30 s PAM-treated group were simultaneously loaded into microfluidic chips and observed continuously for 24 h. The results showed that the cell migration distance in the control group was 186 µm, while that in the 30 s PAM treatment group was 64 µm (Figure 4). The results showed that the migration ability of NSCLC cells in the 30 s PAM treatment group was significantly inhibited as compared with the control group. Therefore, we believe that short-term PAM treatment promoted the migration of NSCLC cells to a certain extent, while long-term PAM treatment significantly inhibited the migration of NSCLC cells.

### 3.3. Effects of PAM Treatment on Intracellular and Extracellular Active Substances

PAM was obtained from the cell culture medium after LTP treatment, and the active substance content was immediately detected. The results are shown in Figure 5; extracellular ROS and RNS increased in a dose-dependent manner with the prolongation of treatment time. After 4 h of incubation, the extracellular ROS decreased significantly, but the extracellular RNS did not significantly change. In addition, the brightness of the green fluorescence representing intracellular ROS content also increased in a dose-dependent manner with treatment time 4 h after LTP treatment (Figure 6). The results showed that, after LTP treatment, intracellular ROS and extracellular RONS increased in a dose-dependent manner. Moreover, the decreased extracellular ROS content and increased intracellular ROS content at 4 h after LTP treatment further indicated that high-dose ROS produced by LTP might enter NSCLC cells as messengers and change the biological activities of cells to a certain extent, for example, inhibiting the migration of NSCLC cells.

To further verify that the inhibition of migration of NSCLC cells by PAM was due to the increase in ROS production caused by LTP treatment, a further set of ROS verification experiments was performed using PC-9 cells. Compared with the control group, there was no statistical difference in the area of cell scratches (Figure 7A,B) and the number of cells passing through Transwell in each group (Figure 7C,D) pretreated with NAC, as shown in Figure 7. The inhibition of cell migration caused by PAM was eliminated with the addition of NAC. The evidence suggested that the migration inhibition of NSCLC cells by PAM was due to the increase in ROS induced by LTP.

### 3.4. Reduction in Mitochondrial Membrane Potential in NSCLC Cells Treated with PAM 

Fluorescence microscopy was used to observe the experimental results. As shown in Figure 8, the green/red fluorescence ratio increased in a dose-dependent manner with the prolongation of LTP treatment time. Compared with the control group, the green/red ratio of H460 cells in the long-term PAM treatment group was 10.04 times that of the control group. The green/red ratio in the PC-9 cell long-term PAM treatment group was 25.76 times that of the control group. The results showed that mitochondrial membrane potential in NSCLC cells decreased in a dose-dependent manner with the prolongation of LTP treatment. Long-term LTP treatment significantly reduced the mitochondrial membrane potential of cancer cells and induced early apoptosis. 

### 3.5. Effect of PAM Treatment on the Cell Cycle of NSCLC Cells

Flow cytometry was used to detect the changes in the division cycle of PC-9 cells after LTP treatment for different durations. The results are presented in Figure 9 and shown a slight increase (46.9%) in the number of cancer cells in the G0/G1 phase in the 10 s PAM treatment group compared with the control group (43.7%). With further LTP treatment, the number of cells in the G2/M phase increased in a dose-dependent manner and peaked at 30 s in the PAM treatment group (33.1%). The results showed that short-term treatment with PAM might slightly promote the proliferation of NSCLC cells, while long-term treatment with PAM could block PC-9 cells in the G2/M phase and inhibit the division and proliferation of cancer cells. 

### 3.6. PAM Treatment Inhibited the EMT Process

The EMT process was closely related to cell migration. Western blot was used to detect the expression levels of EMT-process-related proteins in NSCLC cells after LTP treatment. The results are shown in Figure 10. In the 10 s PAM treatment group, the expressions of N-Cadherin and Vimentin increased, while the expression of E-Cadherin decreased. However, with the further prolongation of PAM treatment, the expression levels of N-Cadherin and Vimentin decreased in a dose-dependent manner (20 s and 30 s), and the expression level of E-Cadherin increased in a dose-dependent manner. The evidence suggested that prolonged LTP treatment could inhibit the migration of NSCLC cells by inhibiting the EMT process. 

### 3.7. PAM Inhibits the Migration of NSCLC Cells through the Wnt/β-Catenin and VEGF/VEGFR Pathways

Compared with the control group, the expressions of Wnt3A and *β*-catenin as well as their downstream proteins c-Jun and Cyclin D1 slightly increased in the 10 s LTP treatment group. However, expression levels of Wnt3A and *β*-catenin and their downstream proteins decreased in a dose-dependent manner in the 20 s and 30 s PAM-treated groups with the prolongation of LTP treatment (Figure 11). These results indicated that short-term PAM treatment might promote the migration of NSCLC cells by activating the Wnt/*β*-catenin signaling pathway, while long-term PAM treatment could inhibit the migration of cancer cells by inhibiting this signaling pathway.

In addition, WB experiments were also used to analyze the VEGF/VEGFR pathway. Compared with the control group, the expressions of vascular endothelial growth factor (VEGF) and vascular endothelial growth factor receptor (VEGFR) proteins in the short-term PAM treatment group increased, while the expressions of VEGF and VEGFR proteins decreased in a dose-dependent manner when the LTP treatment time further increased (Figure 11). These results indicated that short-term treatment with PAM might promote the expression of the angiogenic gene in NSCLC cells, while long-term PAM treatment could inhibit its expression.

We also treated PC-9 cells with an inhibitor (IWP-2) of the Wnt/*β*-catenin signaling pathway to explore whether PAM-induced inhibition of NSCLC cell migration was mediated by the inhibition of the Wnt/*β*-catenin signaling pathway. As shown in Figure 12, a significant decrease in cell migration was observed in the 30 s PAM-treated group in the Transwell experiment when compared to the control group. Compared with the 30 s PAM treatment group, the inhibition of cell migration in the IWP-2 + PAM treatment group was reversed to some extent, and the number of migrated cells significantly increased. In addition, there was no significant difference in the number of cell migrations between the IWP-2 and PAM + IWP-2 groups. The results demonstrated that long-term PAM treatment inhibited the migration of NSCLC cells through inhibiting the Wnt/*β*-catenin signaling pathway.

## 4. Discussion

The blood supply of human lung organs is extremely rich, and tumor cells from lung tumors can transfer to various parts of the body with the blood flow, forming near-distant metastatic tumors. Studies have shown that PAM can not only inhibit the migration of prostate cancer cells, block the cell cycle, and induce apoptosis [44], it can also inhibit the activity of breast cancer cells and their migration ability [45]. Studies have shown that high levels of ROS have been proved to be an effective cancer cell inhibitor, which can lead to oxidative stress and damage cellular biomolecules (including lipids, protein, and DNA) [46,47]. Kim et al. proved that LTP treatment at different times led to an increase in intracellular ROS in different degrees, and induced the apoptosis and even necrosis of lung cancer A549 cells [26]. In addition, as a natural polyphenol, STAT3 can be degraded by ROS-dependent proteasome to inhibit the migration, metastasis, and tumor growth of breast cancer [47,48]. Other studies have shown that the mitochondrial state of cells is closely related to the intracellular ROS level [49]. The increase in ROS level can lead to the decrease in mitochondrial membrane potential and mitochondrial dysfunction, thus inhibiting the growth and migration of renal clear cell cancer cells [50]. In addition, LTP can induce the early apoptosis of rat retinoblastoma cells by increasing the production of mitochondrial ROS, reducing mitochondrial membrane potential, and swelling mitochondria [51]. PAM treated with LTP can also inhibit the proliferation of osteosarcoma cells and induce the apoptosis of osteosarcoma cells by increasing the intracellular ROS level and decreasing the mitochondrial membrane potential [52]. In addition, high levels of ROS can also block the cell division cycle. Studies have shown that PAM obtained after long-term LTP treatment can arrest cells in the G2/M phase, resulting in cell cycle arrest and the apoptosis of cholangiocarcinoma cells and soft tissue sarcoma cells [53,54]. In addition, Lai et al. found in their research that acrylamide can stimulate NSCLC cells to produce excessive ROS, thus blocking the cell division cycle in the G2/M phase [55]. Sugiol can promote the accumulation of ROS in pancreatic cancer cells, leading to the stagnation of cell division in the G2/M phase, thus inhibiting the migration ability of cancer cells [56]. These studies show that high levels of ROS in cells can be used as messengers to mediate the migration of cancer cells. Therefore, in this experiment, we have reason to think that the high level of ROS produced through long-term LTP treatment entered NSCLC cancer cells as messengers, which reduced the mitochondrial membrane potential, destroyed the mitochondrial function, arrested its division cycle in the G2/M phase, lowered the vitality of NSCLC cells, and then inhibited their proliferation and migration ability.

EMT is an important process for cancer cells to acquire invasiveness [57]. When cancer cells lose epithelial characteristics and gain interstitial characteristics, their intercellular adhesion ability is weakened, their migration ability is enhanced, and their migration and invasion ability are significantly enhanced [58]. Zhou et al. have shown that mitochondrial ROS is the key factor of EMT in alveolar epithelial cells under hypoxia [59]. In addition, Yuan et al. found that Cucurbitacin B can inhibit the migration of A549 cells in vitro by inducing ROS accumulation in cells [60]. 

In this study, with the prolongation of LTP treatment time, the intracellular ROS level increased in a dose-dependent manner. Therefore, we think that PAM obtained via long-term LTP treatment can inhibit the EMT and migration of NSCLC cells by increasing the ROS levels entering cells. Another study shows that the typical Wnt signaling pathway is considered the driving factor of colon cancer. As a functional effector of the Wnt signaling pathway, the modification and degradation of *β*-catenin is a key event in the Wnt signaling pathway and the occurrence and development of colon cancer [61]. At the same time, tumor cells can accelerate the EMT process by activating the Wnt/*β*-catenin signaling pathway, thus completing distant metastasis [62]. Studies have shown that glutathione peroxidase 2 (GPX2) in cervical cancer cells activates EMT through the Wnt/*β*-catenin signaling pathway, leading to poor tumor prognosis, and confirmed that GPX2 can reduce apoptosis by reducing intracellular ROS accumulation, which fully proved the close relationship between ROS and the Wnt/*β*-catenin signaling pathway and EMT process in tumor cells [63]. At the same time, the over-expression of Wnt3A (Wnt is the representative of the Wnt ligand) can change the cell morphology by stimulating the Wnt/*β*-catenin signaling pathway, regulating the expression of EMT-related proteins, and accelerating the metastasis and progress of colon cancer [64]. Therefore, it is reasonable to think that the high level of ROS produced through PAM treatment after long-term LTP treatment inhibits the EMT process of NSCLC cells, thus inhibiting their migration through the Wnt/*β*-catenin signaling pathway. Angiogenesis is a complex dynamic process, which is a basic event in the process of tumor growth, metastasis, and spread. The VEGF pathway is considered one of the key regulatory factors in this process [65]. Among them, the VEGF receptor VEGFR 2 plays an important role in signal-transduction-regulating cell proliferation and migration under physiological or pathological conditions [66]. Reducing the high level of angiogenic factors secreted by tumor cells and normalizing tumor angiogenesis have become new tumor treatment strategies [67]. Studies have also shown that ROS derived from intracellular nicotinamide adenine dinucleotide phosphate (NADPH) oxidase is very important for VEGF signal transduction in vitro and angiogenesis in vivo [68]. Therefore, in our experiment, we believe that PAM produced via long-term LTP treatment inhibits the expression of angiogenic factors in NSCLC cells by producing high levels of ROS, thus inhibiting the angiogenesis of NSCLC, thus inhibiting the occurrence and development of cancer.

The results of the MTT assay showed that different NSCLC cell lines had different sensitivities to PAM. Studies have shown that the effect of PAM on NSCLC cells is mainly mediated by increasing the intracellular ROS level. ROS first contacts the cell membrane, so the research on PAM-sensitive types mainly focuses on the cell membrane. The first view is that the content of aquaporin on the cell surface is the key to the sensitivity of cells to LTP. The higher the content of aquaporin on the cell surface, the more ROS will enter the cell and the greater the damage to the cell [69]. The second view is that the content of oxidoreductase on the surface of the cell membrane can amplify the sensitivity of PAM to cells [70]. The last view is that a low cholesterol level will increase oxidation and cavity formation and increase the sensitivity of cells to PAM [71]. In addition, because the data obtained from the experimental results of PC-9 cells are more stable, significant, and representative, the following experiments mainly focus on the influence of PAM on PC-9 cells. Studies have shown that cells show different cell states and fates to different concentrations of H_2_O_2_ (the main species of reactive oxygen species) [72]. For nerve cells, at levels below 1nM H_2_O_2_, cell growth is slow and development and regeneration are impaired; in addition, the cells tended to be in a resting state with no tendency to proliferate and differentiate. When the nerve cells were exposed to H_2_O_2_ in the range of 1–10 nM, the growth of axons and dendrites was promoted. It was mainly because the cells produced oxidative stress under this H_2_O_2_ concentration, which stimulated the proliferation and differentiation of cells. In addition, the study also showed that a modest increase in H_2_O_2_ concentration (up to about 100 nM) could further promote dendritic growth. Abnormally high H_2_O_2_ (more than 100 nM) can lead to the death of nerve cells and tissue degradation [72].

According to the relationship between different concentrations of H_2_O_2_ and the state and fate of nerve cells, we believe that this characteristic may exist in all cells, including NSCLC tumor cells. There are a lot of studies on long-term LTP treatment inducing tumor cell apoptosis to kill tumors [40,73], and there are also many studies showing that short-term LTP treatment can accelerate wound healing by promoting the proliferation and migration of normal skin cells [74,75,76]. The different biological effects of these cells on LTP treatment depend on the length of LTP treatment time. As discussed in this paper, we believe that the different biological effects of these cells on LTP treatment are caused by different concentrations of ROS produced through LTP treatment for different lengths of time. Our experimental results show that PAM treated with LTP for a short time (10 s) promotes the migration of PC-9 cells, while PAM treated with LTP for a long time (30 s) obviously inhibits its migration. We think that this result is also caused by different concentrations of ROS produced by different LTP treatment times in entering the cells, which makes the cells produce different degrees of oxidative stress. The low concentration of ROS entering the cells stimulates the cells to produce a certain degree of oxidative stress, and the cells produce favorable exciting effects. However, when the concentration of ROS entering cells exceeds the tolerance range of the cells, it will bring adverse biological effects to the cells. However, the experimental object of this paper is NSCLC tumor cells that endanger human health. Therefore, the inhibitory effect of high-concentration ROS produced by LTP on cell migration is exactly what we expected, which is beneficial to the treatment of clinical tumors and the prognosis of patients, and provides some theoretical support for the future clinical treatment of NSCLC with LTP.

According to the results of our experiment, PAM treated with LTP for a long time (30 s) can obviously inhibit the migration of NSCLC cells through the Wnt/*β*-catenin cell pathway, which makes PAM very likely to be an inhibitor of NSCLC tumor metastasis in the future. Our plan and assumption is to provide the tumor site with PAM in a certain way (such as injection to the tumor site), especially for patients with advanced NSCLC. By inhibiting the metastasis of tumor cells, we can buy time for the treatment of tumors (such as surgical resection) or improve the prognosis of patients. At the same time, we will further explore the specific molecular substances in PAM that inhibit tumor cell migration, so as to further concentrate and remove unnecessary impurities, so as to achieve a single-cell migration inhibition effect on tumor cells. Of course, we still have a lot of work to undertake before PAM enters clinical application, for example, the further exploration of whether PAM has the same inhibitory effect on the NSCLC cells of p53mut. In addition, we also need to conduct many animal experiments and, later, clinical experiments. Through a lot of data analysis and clinical experimentation, PAM can finally be better used in the clinical setting.

## 5. Conclusions

This study was the first to report the effects of PAM on the migration ability of NSCLC cells. Our results indicated that extracellular RONS and intracellular ROS levels in NSCLC increased in a dose-dependent manner with the increase in LTP treatment. The viability of NSCLC cells was controlled at about 75% by the MTT assay to ensure that the inhibitory effect of PAM on cancer cell migration was not caused by decreased cell viability. The results of the scratch test, cell migration test using the Transwell tool, and the cell migration detection test using microfluidic chips mimicking an in vivo environment all showed that short-term PAM treatment promoted the cell migration of NSCLC, while long-term PAM treatment significantly inhibited its migration. Further exploration is required in the revealed internal mechanism where the mitochondrial membrane potential of NSCLC cells decreased with the increase in LTP treatment. Cell cycle detection also showed that long-term PAM treatment could block the cells in the G2/M phase. The results showed that long-term treatment with PAM induced the early apoptosis of NSCLC cells by reducing the mitochondrial membrane potential and blocking the cell cycle. In addition, the WB assay results showed that long-term PAM treatment inhibited the cell migration of cancer cells in the Wnt/*β*-catenin pathway, which inhibited the EMT process. Therefore, this study showed that high ROS generated by long-term PAM discharge could reverse the EMT process by inhibiting the Wnt/*β*-catenin pathway, thereby inhibiting the migration of NSCLC cells. The results of this study provided strong theoretical support for the clinical treatment of NSCLC with PAM.

## Figures and Tables

**Figure 1 biomolecules-13-01073-f001:**
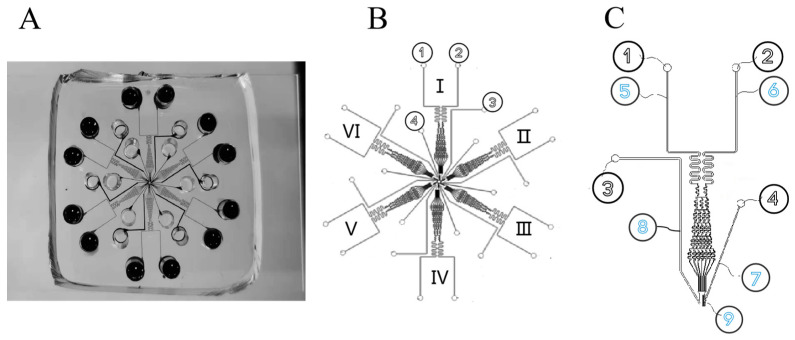
Schematic diagram of microfluidic chip structure. (**A**) Physical diagram of microfluidic chip. (**B**) Schematic diagram of 6-channel chip. The numbers I- VI represent that this chip is a six-channel chip. Among them, I, II and III channels were the control group, and IV, V and VI channels were the 30 s experimental group. (**C**) Single-channel loading unit. ① Chemokine injection port. ② Cell culture solution injection port. ③ Waste liquid outlet. ④ Cell injection port. ⑤ Chemokine transport channel. ⑥ Cell culture medium delivery channel. ⑦ Cell transmission channel. ⑧ Waste liquid channel. ⑨ Cell isolation zone.

**Figure 2 biomolecules-13-01073-f002:**
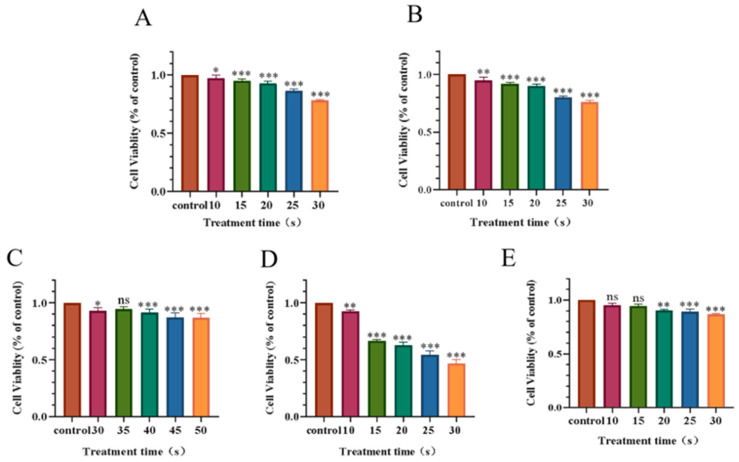
Effect of PAM with different treatment time on NSCLC cell viability after incubation for 24 h. (**A**) Cell viability of H460. (**B**) Cell viability of PC-9. (**C**) Cell viability of A549. (**D**) Cell viability of H1299. (**E**) Cell viability of H1975. Data represent the mean ± SD of three independent experiments. “ns” means no statistical difference. * *p* < 0.05, ** *p* < 0.01,*** *p* < 0.001 with ANOVA compared with the control.

**Figure 3 biomolecules-13-01073-f003:**
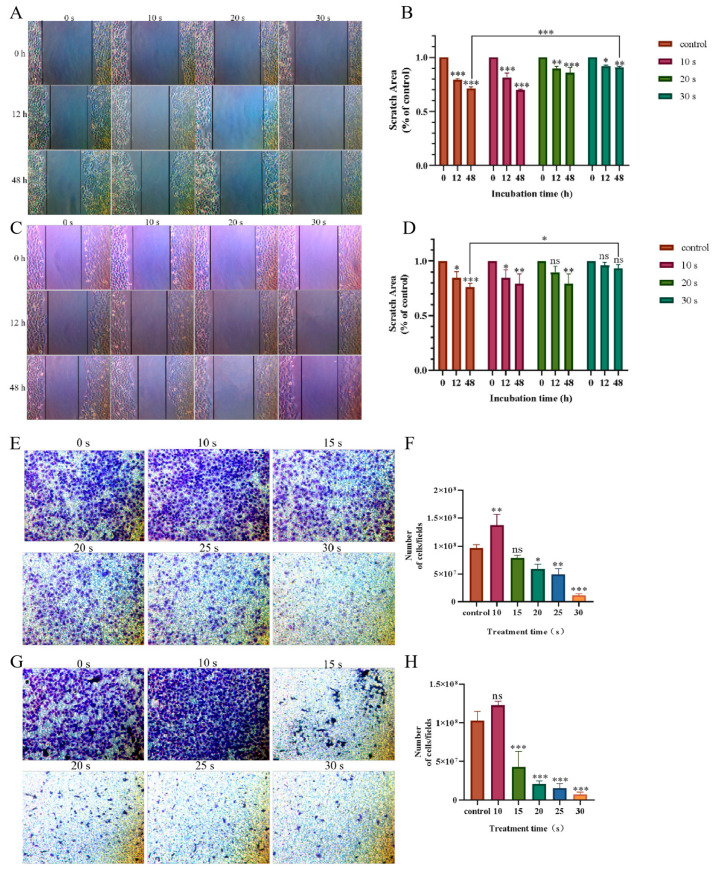
Effect of PAM incubation on the migration ability of NSCLC cells. (**A**) Wound healing experiments were conducted to detect the effect of PAM incubation on the migration of H460 cells. (**B**) Quantification of the scratch area of the H460 cells after PAM treatment. (**C**) Wound healing experiments were conducted to detect the effects of PC-9 cells incubated with PAM on their migration. (**D**) Quantification of the scratch area of the PC-9 cells after PAM treatment. (**E**) Transwell cells were used to analyze the effect of H460 cells incubated with PAM on their migration capacity. (**F**) The number of H460 cells passing through the Transwell chamber after PAM incubation. (**G**) Transwell cells were used to detect the effect of PAM incubation on PC-9 cell migration. (**H**) The number of PC-9 cells passing through the Transwell chamber after PAM incubation. Data represent the mean ± SD of three independent experiments. “ns” means no statistical difference. * *p* < 0.05, ** *p* < 0.01,*** *p* < 0.001 with ANOVA compared with the control.

**Figure 4 biomolecules-13-01073-f004:**
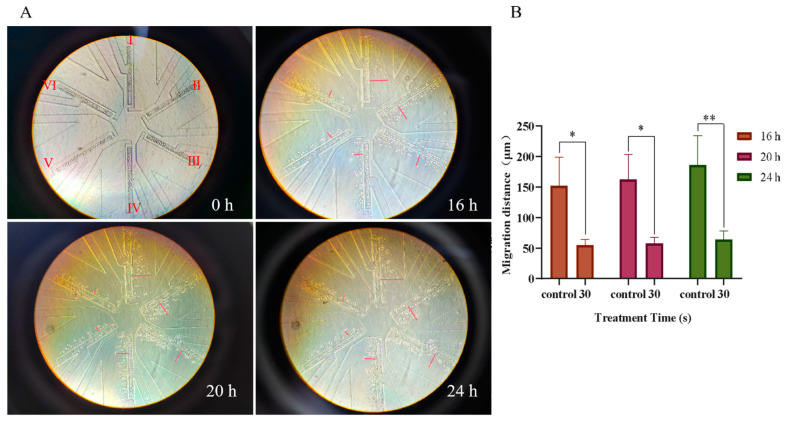
Observation of the effects of PAM treatment on PC-9 cell migration using high throughput microarray. (**A**) The migration of PC-9 cells on the chip was recorded at regular intervals, and the migration locations of cells on the chip at 0 h, 16 h, 20 h, and 24 h were recorded. The Ⅰ, Ⅱ, and Ⅲ channels were used as the control group, and the Ⅳ, Ⅴ, and Ⅵ channels were used as the LTP (30 s) treatment group. The auxiliary red line in the figure represents the straight-line distance of PC-9 cell migration. (**B**) Quantification of the migration distance of PC-9 cells incubated at different times after 30 s PAM treatment. Data represent the mean ± SD of three independent experiments. * *p* < 0.05, ** *p* < 0.01 with ANOVA compared with the control.

**Figure 5 biomolecules-13-01073-f005:**
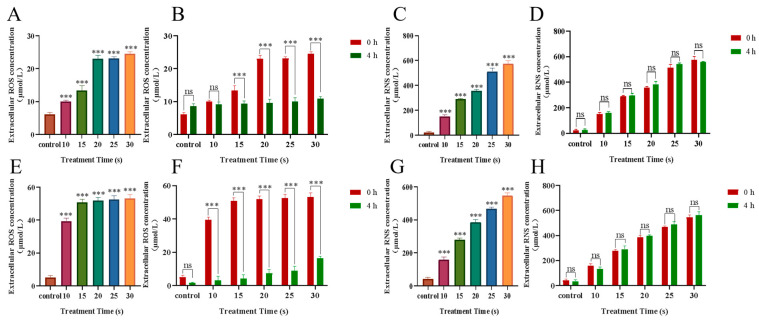
Extracellular H_2_O_2_ and NO levels. (**A**) H460 cells’ extracellular H_2_O_2_ concentration at 0 h after treatment with PAM. (**B**) H460 cells’ extracellular H_2_O_2_ concentration at 0 h and after incubation for 4 h after treatment with PAM. (**C**) H460 cells’ extracellular NO concentration at 0 h after treatment with PAM. (**D**) H460 cells’ extracellular NO concentration at 0 h and after incubation for 4 h after treatment with PAM. (**E**) PC-9 cells’ extracellular H_2_O_2_ concentration at 0 h after treatment with PAM. (**F**) PC-9 cells’ extracellular H_2_O_2_ concentration at 0 h and after incubation for 4 h after treatment with PAM. (**G**) PC-9 cells’ extracellular NO concentration at 0 h after treatment with PAM. (**H**) PC-9 cells’ extracellular NO concentration at 0 h and after incubation for 4 h after treatment with PAM. Data represent the mean ± SD of three independent experiments. “ns” means no statistical difference. *** *p* < 0.001 with ANOVA compared with the control.

**Figure 6 biomolecules-13-01073-f006:**
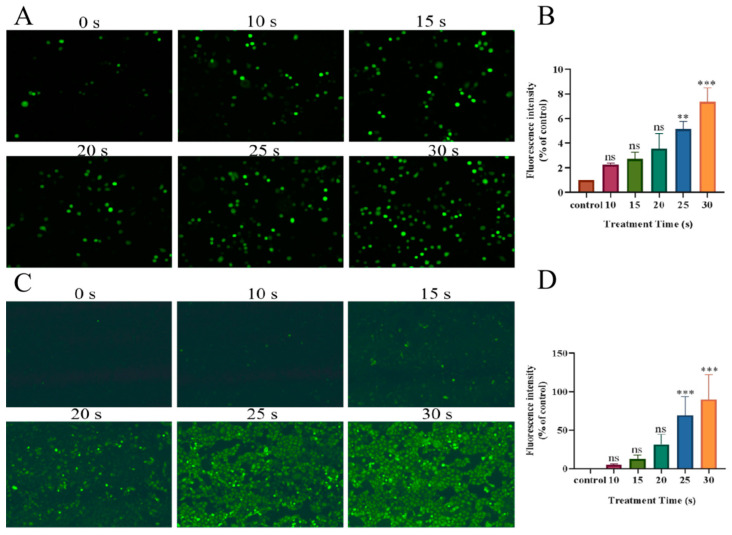
After PAM treatment, green fluorescence represents the level of ROS in H460 cells and PC-9 cells. (**A**) Fluorescence image of ROS level in H460 cells after PAM treatment. (**B**) H460 cells’ intracellular fluorescence level quantification. (**C**) Fluorescence image of ROS level in PC-9 cells after PAM treatment. (**D**) PC-9 cells’ intracellular fluorescence level quantification. In (**B**,**D**), brown bars, red bars, yellow-green bars, green bars, blue bars and yellow bars represent the control group, 10 s PAM treatment group, 15 s PAM treatment group, 20 s PAM treatment group, 25 s PAM treatment group and 30 s PAM treatment group. Data represent the mean ± SD of three independent experiments. “ns” means no statistical difference. ** *p* < 0.01,*** *p* < 0.001 with ANOVA compared with the control.

**Figure 7 biomolecules-13-01073-f007:**
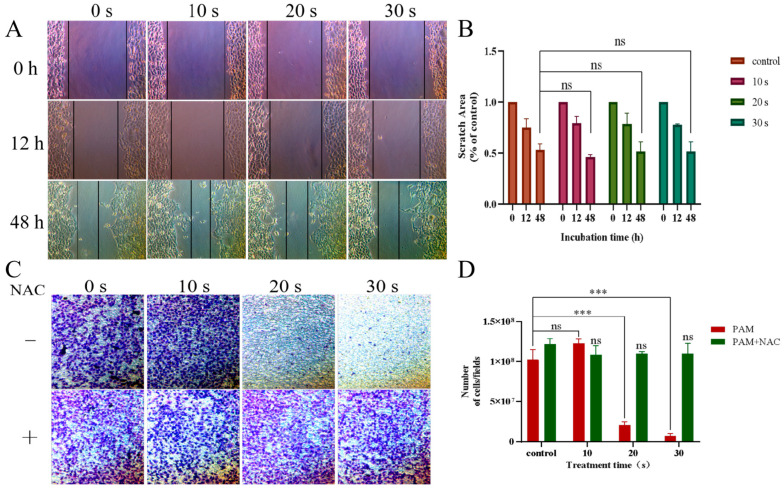
PAM-inhibited PC-9 cell migration via ROS. (**A**) After NAC-pretreated PC-9 cells, the cells were PAM-incubated for 0, 12, and 48 h and the migration image was scratched. (**B**) Quantification of the scratch migration in PC-9 cells after treatment with PAM and incubation for 0 h, 12 h, and 48 h, respectively. (**C**) Transwell image of PC-9 cells before PAM treatment with or without NAC pretreatment. (**D**) Quantification of Transwell results in PC-9 cells with and without NAC pretreatment before PAM treatment. Data represent the mean ± SD of three independent experiments. “ns” means no statistical difference. *** *p* < 0.001 with ANOVA compared with the control.

**Figure 8 biomolecules-13-01073-f008:**
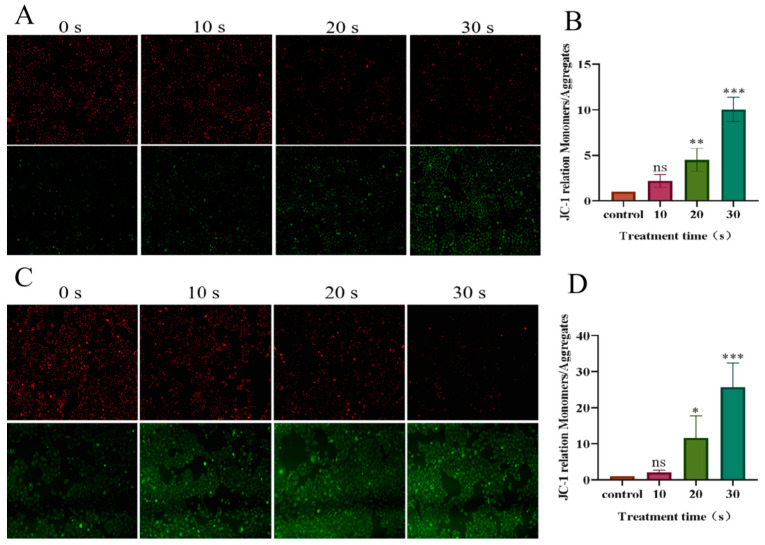
Effect of PAM treatment on mitochondrial membrane potential in PC-9 and H460 cells. JC-1 forms aggregates with red fluorescence (in healthy mitochondria). With the decrease of membrane potential, JC-1 became a monomer and showed green fluorescence.(**A**) Fluorescence image of mitochondrial membrane potential in H460 cells after PAM treatment. (**B**) Fluorescence quantification of mitochondrial membrane potential in H460 cells after PAM treatment. (**C**) Fluorescence image of mitochondrial membrane potential in PC-9 cells after PAM treatment. (**D**) Fluorescence quantification of mitochondrial membrane potential in PC-9 cells after PAM treatment. Data represent the mean ± SD of three independent experiments. “ns” means no statistical difference. * *p* < 0.05, ** *p* < 0.01, *** *p* < 0.001 with ANOVA compared with the control.

**Figure 9 biomolecules-13-01073-f009:**
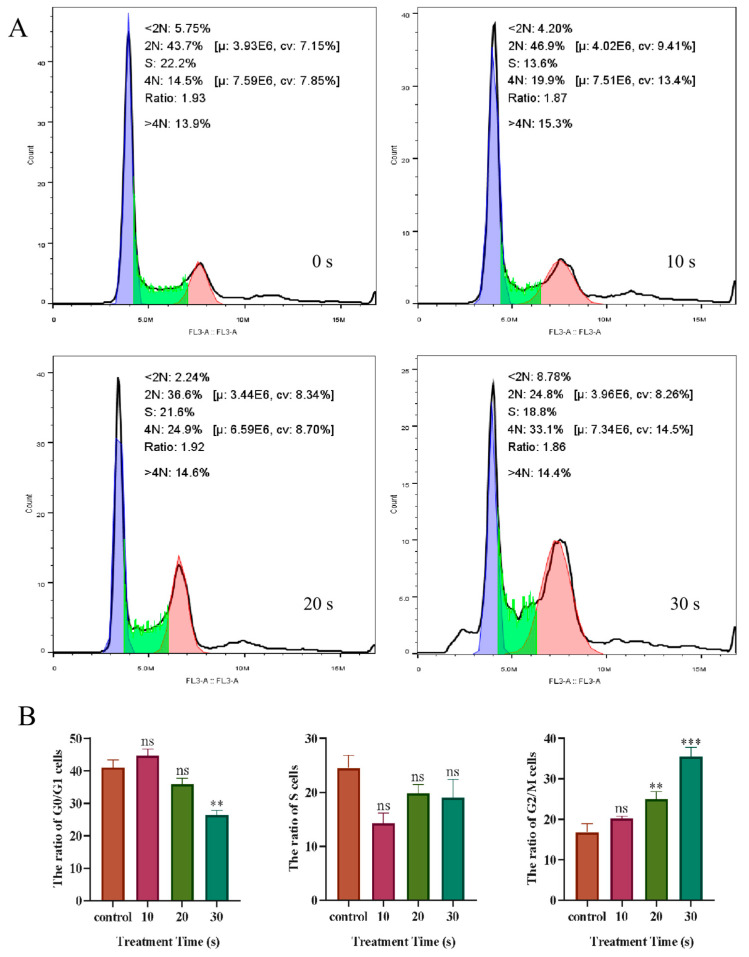
The effect of LTP treatment on the PC-9 cell cycle. (**A**) Flow cytometry was used to analyze the cycle distribution of LTP-treated PC-9 cells. Among them, the purple region represents G0/G1 phase, the green region represents S phase and the red region represents G2/M phase. (**B**) Quantitation of the percentage of cells in the G0/G1, S, and G2/M periods after LTP treatment. Data represent the mean ± SD of three independent experiments. “ns” means no statistical difference. ** *p* < 0.01, *** *p* < 0.001 with ANOVA compared with the control.

**Figure 10 biomolecules-13-01073-f010:**
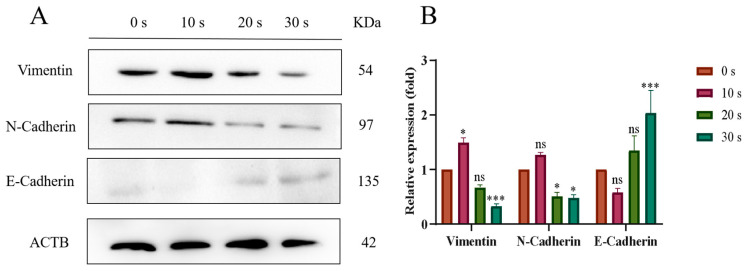
Effect of LTP treatment on the expression of EMT-related proteins in PC-9 cells. (**A**) WB bands of Vimentin, N-Cadherin, and E-Cadherin proteins. (**B**) Quantitative analysis of protein expression levels. Among them, brown bars, red bars, yellow-green bars and green bars represent the control group, PAM treatment for 10 s group, PAM treatment for 20 s group and PAM treatment for 30 s group respectively. Data represent the mean ± SD of three independent experiments. “ns” means no statistical difference. * *p* < 0.05, *** *p* < 0.001 with ANOVA compared with the control.

**Figure 11 biomolecules-13-01073-f011:**
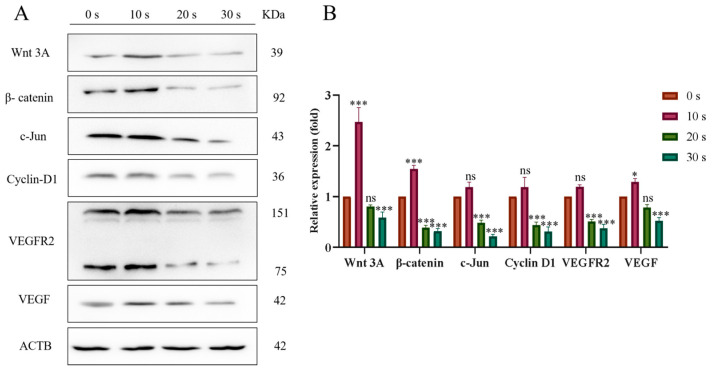
Effects of LTP treatment on the expression of Wnt/*β*-catenin and VEGF-signaling-pathway-associated proteins in PC-9 cells. (**A**) WB bands of Wnt3A, *β*-Catenin, Cyclin D1, c-Jun, VEGFR2, and VEGF proteins. (**B**) Quantitative analysis of each protein expression level. Data represent the mean ± SD of three independent experiments. “ns” means no statistical difference. * *p* < 0.05, *** *p* < 0.001 with ANOVA compared with the control.

**Figure 12 biomolecules-13-01073-f012:**
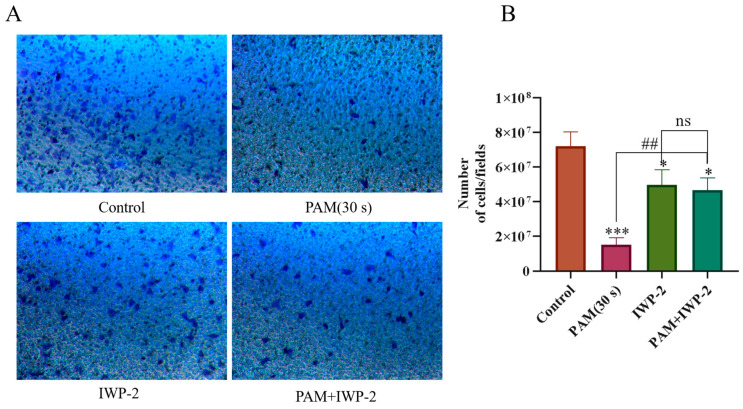
PAM treatment inhibited PC-9 cell migration through the Wnt/*β*-catenin signaling pathway. (**A**) Transwell migrated images from the control group, the PAM (30 s) treatment group, the IWP-2 alone treatment group, and the IWP-2 pretreatment group. (**B**) Quantitative analysis of cell migration. Data represent the mean ± SD of three independent experiments. “ns” means no statistical difference. * *p* < 0.05, *** *p* < 0.001 with ANOVA compared with the control. ## *p* < 0.01.

## Data Availability

Not applicable.

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
