# Peer review of "Low-Temperature Plasma-Activated Medium Inhibits the Migration of Non-Small Cell Lung Cancer Cells via the Wnt/β-Catenin Pathway"

_biomolecules, 2023, doi:10.3390/biom13071073_

Round 1

Reviewer 1 Report

The article Zhang et al is devoted to the plasma activation medium that inhibit the migration ability of NSCLC cells. There are a lot of interesting results, however authors are focused on the cell lines that expressed p53wt. It will be interesting also to understand is this effects p53 dependent or not.

The Quality of English is good

Author Response

Reply: Thank you very much for your comments and suggestions. We are very interested in your suggestion and are very eager to explore your questions. However, our laboratory conditions are temporarily limited, and we don't know enough about the professional knowledge in this field. Therefore, in view of the current conditions, we may not be able to explore the effect of plasma activating medium (PAM) on p53mut NSCLC cells in a short time. Of course, according to your suggestion, we will further explore whether the influence of PAM on NSCLC cells depends on p53 by upgrading the conditions and facilities of our laboratory and improving our mastery of professional knowledge in this field in future experiments. Thank you again for your comments on our experimental research.

Reviewer 2 Report

The manuscript titled: „Low temperature plasma-activatied medium inhibits the migration of NSCLC cells via Wntβ-catenin pathway” focuses on new approach to reduce non small cell lung cancer (NSCLC) migration via inhibition of WNT/B-catenin pathway by ROS induced by low temperature plasma. Lung cancer is considered as the leading cause of cancer related deaths, and one of THE most prominent death cause (at all) worldwide. Currently, surgical resection of lung cancer tumor with appropriate margin is best treatment modality, however not every tumor might be resected, mainly due to late diagnosis, thus different forms of therapy such as chemotherapy, targeted therapy or radiotherapy are commonly used.

Authors of this manuscript tested the concept of applying medium containing high level of ROS generated by low temperature plasma to NSCLC cells, thus influencing WNT/B-catenin pathway, and in aftermath reducing migratory and invasive abilities of tested cells. This new approach, may be clinically beneficial leading to decreased metastasis.

Manuscript contains substantial amount of data, presented on 12 figures, presented in rather clear manner. However I have some comments regarding this manuscript (divided into minor and major comments) that are listed below:

Major comment:

In my opinion, this manuscript requires proper English edition (especially in case of extensive overuse of past tens referring to general subjects and current knowledge).

In materials and methods section, authors points that T-test was used for data analysis, whereas ANOVA is  listed in majority of figures. Additionally no indication of data distribution is presented, thus it is unknown whether t-test is appropriate statistical tool. I suggest using Shapiro Wilk test to analyze distribution.

It is not precise whether authors calculate enclosure of wound (in wound healing assay) by “scratch” surface area or distance between cells that actually migrate towards each other. Please change the description of the method.

MTT assay, why cells were not seeded in equal amounts, but authors rather relay on confluence (80%) that is (in my opinion) rather inconvenient, as different cells presents range of sizes. Beside, confluence is hard to be measured adequately and to be repeated on every experiment.

According to the Biomolecules guidance for authors Article should consist of: Introduction, Materials and Methods, Results, Discussion, Conclusions (optional), Patents. This manuscript combines Results with Discussion into one paragraph, and in my opinion this decision leads to a poor result discussion  with omission of some key findings. In my opinion differences between 10s and 30s LTP exposure medium effect on NSCLC cells should be deeply discussed, as they present opposite results. Judging on provided western blots I am not convinced that Vimentin (in particular) and N-Cadherin expression presented on fig. 10 are not statistically significant for 10s PAM group. Moreover this effect corresponds with increased migration and Wnt/B-catenin level of this group. Why microfluidic chip migration was not tested on 10s PAM group? Additionally, 10s PAM group presents highest viability, whereas 30s PAM group viability was 75-78% for both PC-9 and H460. This effect might have influence cell migration and invasion. Why authors decided not to use A549 that presents higher viability even in 40-50s PAM? Thus, I strongly encourage authors to deeply analyze and discuss this subject.

I miss section where authors would discuss how this knowledge might be clinically utilized.

Minor comments:

Spelling mistakes (to name a few)

Line 2 (title): “activatied” (it is repeated on many instances) should be activated

Line 6: “Tcchnology” should be Technology

Line 156 “in-cubated” should be incubated

Line 132: “8µL 24-hole” should be 8µm 24-well

Line 291 “Cell viability of NSCLC cells after 24 h incubation with different doses of PAM” – as far as I understand dose is the same, the only difference is medium exposure time for LTP. Please consider changing it.

Fig.3 : ” Statistical method of T-test was used to quantify the scratch area (…)” “The statistical method of T- test was used to quantify the number of cells passing through (…)” – T-test may be used to verify statistical significance of obtained results, but not to quantify area or number of passing cells. Additionally p value significance was pointed to by calculated by ANOVA test. This phrases/paragraph should be rewritten.

Fig. 11 significance is given for: ** p<0.01, ** p<0.001 whereas on the figure 11B only * and ***  are depicted, with no ** (VEGF bars). Please change it.

In my opinion, this manuscript requires proper English edition (especially in case of extensive overuse of past tens referring to general subjects and current knowledge).

Spelling mistakes (to name a few)

Line 2 (title): “activatied” (it is repeated on many instances) should be activated

Line 6: “Tcchnology” should be Technology

Line 156 “in-cubated” should be incubated

Author Response

Reviewer: 2

  1. In my opinion, this manuscript requires proper English edition (especially in case of extensive overuse of past tens referring to general subjects and current knowledge).

Reply: We greatly appreciated the reviewer’s detailed comments. We have carefully revised the spelling of words and the tense of sentences in the article, and all the changes are highlighted in blue font.

  1. In materials and methods section, authors points that T-test was used for data analysis, whereas ANOVA is  listed in majority of figures. Additionally no indication of data distribution is presented, thus it is unknown whether t-test is appropriate statistical tool. I suggest using Shapiro Wilk test to analyze distribution.

Reply: Thank you very much for your comments. According to your suggestion, we use Shapiro Wilk test to test the data, and have made some modifications in the Data analysis section of the article, as follows:

Page 7 Line 278-282: The data of assays were launched at least three repeating independent experiments (n = 3) and presented as means ± S.D. Data were determined using Bonferroni’s corrected one-way analysis of variance (ANOVA) and two-way analysis of variance (2 way-ANOVA). Shapiro Wilk test was used to analyze the distribution of data. The statistical significance difference was based on * P < 0.05, ** P < 0.01, *** P < 0.001.

  1. It is not precise whether authors calculate enclosure of wound (in wound healing assay) by “scratch” surface area or distance between cells that actually migrate towards each other. Please change the description of the method.

Reply: We greatly appreciated the reviewer’s detailed comments. It is confirmed that we measure the migration of cells by the surface area of scratches, and it has been re-described in the wound healing assay section of the article materials and methods, as follows:

Page 3 Line 126-132: H460 and PC-9 cells were incubated in 10% fetal bovine serum until the bottom of the plate fused. Discard the original culture medium in the Petri dish, and scrape off the monolayer on the cell fusion surface with a 20 μL white pipette head. The scraped cells and cell fragments were washed with PBS three times, and the cells were incubated in PAM without FBS. Cell migration of PAM after incubation for 4 h, 12 h, 24 h and 48 h was observed and recorded by an inverted microscope (Guangzhou, China), and the scratch surface area after different incubation time was quantified by using Image-J software with different gray values. Finally, the scratch areas of each treatment group were compared with those of control group, and the ratio was calculated.

  1. MTT assay, why cells were not seeded in equal amounts, but authors rather relay on confluence (80%) that is (in my opinion) rather inconvenient, as different cells presents range of sizes. Beside, confluence is hard to be measured adequately and to be repeated on every experiment.

Reply: Thanks for reviewer’s kindly comment and suggestion. When the cells grew to passage, we added culture medium to make the cells into cell suspension, which was evenly transferred to each Petri dish after being blown and mixed, so we had achieved equal inoculation of the cells during passage. And it has been modified in the cell viability experiment of materials and methods in the article, as follows:

Page 3 Line 111-123: 3- (4, 5-Dimethyl-2-thiazolyl) -2, 5-diphenyltetrazole bromide (MTT, Sigma-Aldrich, St. Louis, MO, USA) was used as a cell viability assay. Five NSCLC cell lines, including H460, PC-9, A549, H1299, and H1975. When the cells grew to be suitable for passage, the adherent cells were digested with trypsin and added with cell culture solution to make cell suspension, and then transferred to each Petri dish evenly. When the cells in each dish grew to about 80% of the bottom of the dish, they were incubated with PAM for 24 h. After the PAM was discarded, 1 mL of pre-prepared MTT working solution was added into each dish for continuous incubation for 4 h. The plates were removed from the MTT working solution and an equal volume of dimethyl sulfoxide (DMSO, Biotechnology, Shanghai, China) was added to each plate to dissolve the crystals in the cells. After sufficient shaking, 150 μL of dissolved liquid was added to each well in a 96-well plate, and a microplate reader was used to determine the absorbance at 492 nm for each well. Cell viability was calculated according to a standard curve.

  1. According to the Biomolecules guidance for authors Article should consist of: Introduction, Materials and Methods, Results, Discussion, Conclusions (optional), Patents. This manuscript combines Results with Discussion into one paragraph, and in my opinion this decision leads to a poor result discussion  with omission of some key findings.

Reply: We are very grateful to the reviewers for their opinions. According to your suggestion, we have described the results and the discussion separately, and added the possible reasons for the different sensitivity of different NSCLC cells to PAM, the difference of the effects of PAM 10 s and 30 s treatment groups, and the future clinical application of PAM in the discussion part.

Page 7 Line 283 - Page 15 Line 485: The result part

Page 16 Line 486 - Page 18 Line 626: The discussion part

  1. In my opinion differences between 10 s and 30 s LTP exposure medium effect on NSCLC cells should be deeply discussed, as they present opposite results.

Reply: Thanks for reviewer’s kindly comments.

Page 17 Line 581-Page 18 Line 614: Studies have shown that cells show different cell states and fates to different concentrations of H2O2 (The main species of reactive oxygen species) [1]. For nerve cells, at levels below 1nM H2O2, cell growth is slow and development and regeneration are impaired; In addition, the cells tended to be in a resting state with no tendency to proliferate and differentiate. When the nerve cells were exposed to H2O2 in the range of 1–10 nM, the growth of axons and dendrites was promoted. It was mainly because the cells produced oxidative stress under this H2O2 concentration, which stimulated the proliferation and differentiation of cells. In addition, the study also showed that a modest increase in H2O2 concentration (up to about 100 nM) could further promote dendritic growth. Abnormally high H2O2 (more than 100 nM) can lead to the death of nerve cells and tissue degradation[1].

According to the relationship between different concentrations of H2O2 and the state and fate of nerve cells, we believe that this characteristic may exist in all cells, including NSCLC tumor cells. There are a lot of studies on long-term LTP treatment inducing tumor cell apoptosis to kill tumor [2,3], and there are also many studies showing that short-term LTP treatment can accelerate wound healing by promoting the proliferation and migration of normal skin cells [4-6]. The different biological effects of these cells on LTP treatment depend on the length of LTP treatment time As discussed in this paper, we believe that the different biological effects of these cells on LTP treatment are caused by different concentrations of ROS produced by LTP treatment for different lengths of time. Our experimental results show that PAM treated with LTP for a short time (10 s) promotes the migration of PC-9 cells, while PAM treated with LTP for a long time (30 s) obviously inhibits its migration. We think that this result is also caused by different concentrations of ROS produced by different LTP treatment time entering the cells, which makes the cells produce different degrees of oxidative stress. The low concentration of ROS entering the cell stimulates the cell to produce a certain degree of oxidative stress, and the cell produces favorable exciting effect; However, when the concentration of ROS entering cells exceeds the tolerance range of cells, it will bring adverse biological effects to cells.

However, the experimental object of this paper is NSCLC tumor cells that endanger human health. Therefore, the inhibitory effect of high concentration ROS produced by LTP on cell migration is exactly what we expected, which is beneficial to the treatment of clinical tumors and the prognosis of patients, and provides some theoretical support for the future clinical treatment of NSCLC by LTP.

  • Sies, H.; Jones, D.P. Reactive oxygen species (ROS) as pleiotropic physiological signalling agents. Nat Rev Mol Cell Biol, 2020, 21(7):363-383.
  • Zhou, Y.; Zhang, Y.; Bao, J.; Chen, J.; Song, W. Low Temperature Plasma Suppresses Lung Cancer Cells Growth via VEGF/VEGFR2/RAS/ERK Axis. Molecules (Basel, Switzerland), 2022, 27, 5934.
  • Wang, Y.; Mang, X.; Li, X.; Cai, Z.; Tan, F. Cold atmospheric plasma induces apoptosis in human colon and lung cancer cells through modulating mitochondrial pathway. Frontiers in cell and developmental biology, 2022, 10, 915785.
  • Shi, X. M.; Xu, G. M.; Zhang, G. J.; Liu, J. R.; Wu, Y. M.; Gao, L. G.; Yang, Y.; Chang, Z. S.; Yao, C. W. Low-temperature Plasma Promotes Fibroblast Proliferation in Wound Healing by ROS-activated NF-κB Signaling Pathway. Current medical science, 2018, 38(1), 107–114.
  • Arjunan, K. P.; Clyne, A. M. Non-thermal dielectric barrier discharge plasma induces angiogenesis through reactive oxygen species. Annual International Conference of the IEEE Engineering in Medicine and Biology Society. Annual International Conference, 2011, 2447–2450.
  • Wu, H.; Zhang, Y.; Zhou, Y.; Yan, Z.; Chen, J.; Lu, T.; Song, W. Low-Dose Non-Thermal Atmospheric Plasma Promotes the Proliferation and Migration of Human Normal Skin Cells. Applied Sciences, 2023, 13(5):2866.

  1. Judging on provided western blots I am not convinced that Vimentin (in particular) and N-Cadherin expression presented on fig. 10 are not statistically significant for 10 s PAM group. Moreover this effect corresponds with increased migration and Wnt/β-catenin level of this group.

Reply: We greatly appreciate the reviewer’s comment. We reanalyzed the WB original images of Vimentin and N-Cadherin in Fig.10, and found that compared with the control group, the expression of Vimentin protein in the 10 s PAM group increased and had certain statistical significance, while the expression of N-Cadherin protein increased slightly but did not have statistical significance. The revised picture was as follows:

  1. Why microfluidic chip migration was not tested on 10 s PAM group?

Reply: Thanks for reviewer’s kindly comments. As shown in the figure below, in the previous experiment, we have used microfluidic chips to detect the migration of cells in the 10 s PAM group. It is only because we want to highlight the inhibitory effect of PAM obtained by 30 s LTP treatment on NSCLC cell migration that we did not put this part of the results into this paper. Results As shown in the following figure, the migration distance of cells in 10 s PAM group began to be promoted at 16 h after incubation, but compared with 10 s PAM group, the migration distance of cells in 30 s PAM group was always inhibited. Therefore, we think that PAM incubation with LTP for a short time will promote the migration of NSCLC cells, while PAM incubation with LTP for a long time will inhibit the migration of NSCLC cells.

  1. Additionally, 10 s PAM group presents highest viability, whereas 30s PAM group viability was 75-78% for both PC-9 and H460. This effect might have influence cell migration and invasion. Why authors decided not to use A549 that presents higher viability even in 40-50s PAM? Thus, I strongly encourage authors to deeply analyze and discuss this subject.

Reply: We greatly appreciated the reviewer’s detailed comments. There are two main reasons why we chose PC-9 and H460 cells instead of A549 cells for the follow-up experiment:

First of all, the clinical topic discussed in this paper is the inhibitory effect of PAM produced by LTP treatment on NSCLC tumor cells, so we selected five cell lines of NSCLC tumor cells: A549, H1299, H1975, H460 and PC-9. In the MTT experiment, it was found that the viability of A549 cells was still 94% after 24 h incubation with PAM obtained by LTP treatment for 30 s, and the viability of other cell lines was less than 80%. Secondly, after repeated MTT experiments with A549, it was found that the cell viability was very unstable, and further experiments could not be completed. Therefore, for two reasons, in order to make our follow-up experimental results more representative and convincing, we chose two NSCLC cells, PC-9 and H460, with more stable experimental data.

  1. I miss section where authors would discuss how this knowledge might be clinically utilized.

Reply: Thanks a lot for the reviewer’s comments.

Page 18 Line 615-Page 18 Line 629: According to the results of our experiment, PAM treated with LTP for a long time (30 s) can obviously inhibit the migration of NSCLC cells through Wnt/β-catenin cell pathway, which makes PAM very likely to be an inhibitor of NSCLC tumor metastasis in the future. Our plan and assumption is to give PAM to the tumor site through a certain way (such as injection to the tumor site), especially to the patients with advanced NSCLC. By inhibiting the metastasis of tumor cells, we can buy time for the treatment of tumors (such as surgical resection) or improve the prognosis of patients. At the same time, we will further explore the specific molecular substances in PAM that inhibit tumor cell migration, so as to further concentrate and remove unnecessary impurities, so as to achieve a single cell migration inhibition effect on tumor cells.

Of course, we still have a lot of work to do before PAM enters clinical application. For example, to further explore whether PAM has the same inhibitory effect on NSCLC cells of p53mut. In addition, we also need a lot of animal experiments and later clinical experiments. Through a lot of data analysis and clinical experiments, PAM can finally be better used in clinic.

  1. Spelling mistakes (to name a few)

Line 2 (title): “activatied” (it is repeated on many instances) should be activated

Line 6: “Tcchnology” should be Technology

Line 156 “in-cubated” should be incubated 

Line 132: “8µL 24-hole” should be 8µm 24-well

Line 291 “Cell viability of NSCLC cells after 24 h incubation with different doses of PAM” – as far as I understand dose is the same, the only difference is medium exposure time for LTP. Please consider changing it.

Reply: We greatly appreciated the reviewer’s detailed comments. According to your suggestion, we have made changes in the corresponding part of the article, as follows:

Page 1 Line 2: Low temperature plasma-activated medium inhibits the migration of NSCLC cells via Wnt/β-catenin pathway

Page 1 Line 6: School of medicine, Anhui University of Science and Technology, Huainan 232001, P.R. China

Page 4 Line 160: After the master mold fabrication was completed, polydimethylsiloxane (PDMS) was injected and incubated at 70℃ for 2 h.

Page 3 Line 136: The experiment was carried out in 8 μm 24-hole Transwell plate (Costar, Washington, DC, USA).

Page 7 Line 299: Effect of PAM with different treatment time on NSCLC cell viability after incubation for 24 h.

  1. 3 : “ Statistical method of T-test was used to quantify the scratch area (…)” “The statistical method of T- test was used to quantify the number of cells passing through (…)” – T-test may be used to verify statistical significance of obtained results, but not to quantify area or number of passing cells. Additionally p value significance was pointed to by calculated by ANOVA test. This phrases/paragraph should be rewritten.

Fig. 11 significance is given for: ** p<0.01, ** p<0.001 whereas on the figure 11B only * and ***  are depicted, with no ** (VEGF bars). Please change it.

Reply: Thanks for reviewer’s kindly comments. According to your suggestion, we have made changes in the corresponding part of the article, as follows:

Page 9 Line 329-339: Fig. 3. Effect of PAM incubation on the migration ability of NSCLC cells. (A) The wound healing experiment was conducted to detect the effect of PAM incubation on the migration of H460 cells. (B) Quantification of scratch area of H460 cells after PAM treatment. (C) Wound healing experiments were conducted to detect the effects of PC-9 cells incubated with PAM on their migration. (D) Quantification of scratch area of PC-9 cells after PAM treatment. (E) Transwell cells were used to analyze the effect of H460 cells incubated with PAM on their migration capacity. (F) The number of H460 cells passing through Transwell chamber after PAM incubation. (G) Transwell cells were used to detect the effect of PAM incubation on PC-9 cell migration. (H) The number of PC-9 cells passing through Transwell chamber after PAM incubation. Data represent the mean ± SD of three independent experiments. “ns” means no statistical difference. * p < 0.05, ** p < 0.01,*** p < 0.001 with ANOVA and 2way-ANOVA compared with the control.

Page 15 Line 475-479: Fig. 11. Effects of LTP Treatment on the Expression of Wnt/β-catenin and VEGF Signaling Pathway-associated Proteins in PC-9 Cells (A) WB bands of Wnt3A, β-Catenin, Cyclin D1, c-Jun, VEGFR2 and VEGF proteins. (B) Quantitative analysis of each protein expression level. Data represent the mean ± SD of three independent experiments. “ns” means no statistical difference. * p<0.05, *** p < 0.001 with ANOVA compared with the control, Shapiro Wilk test.

Reviewer 3 Report

In this manuscript, the authors studied the molecular mechanism of low temperature plasma activation medium (PAM) inhibiting the migration ability of NSCLC cells. They used several methods to examine the effect of PAM on the proliferation and migration of lung cancer cell lines (mainly PC-9). Their previous study has demonstrated that the cell growth lung cancer cells (CALU-1 and SPC-A1) are directly suppressed by low temperature plasma (LTP) via VEGF/VEGFR2/RAS/ ERK Axis (Ref. 41: Molecules 2022, 27, 5934. https://doi.org/10.3390/molecules27185934. Low Temperature Plasma Suppresses Lung Cancer Cells Growth via VEGF/VEGFR2/RAS/ERK Axis). In addition, several studies have showed that LTP and PAM can directly affect cell proliferation and migration of lung cancer via reactive oxygen species/reactive nitrogen species and several signal transduction pathways. (For example, Ref. 21: Oxid Med Cell Longev. 2022 Mar 18; 2022: 9014501. doi: 10.1155/2022/9014501. Low-Temperature Plasma-Activated Medium Inhibited Proliferation and Progression of Lung Cancer by Targeting the PI3K/Akt and MAPK Pathways). Therefore, this study only provided that the Wnt/β-catenin pathway may be one of the mechanisms involved in PAM's inhibition of lung cancer cell migration. I have some suggestions for the authors.

Major comment:

1.        They used an inhibitor, IWP-2, to show PAM treatment inhibited PC-9 cell migration through the Wnt/β-catenin signaling pathway. It needs to check whether IWP-2 could really rescue the PAM-inhibited Wnt/β-catenin signaling pathway. In addition, this result needs to be validated using other lung cancer cell lines and methods such as siRNA or shRNA.

2.        The explanation for why only PC-9 cells were used for most experiments. Please provide possible reasons for the different sensitivity of different lung cancer cell lines to PAM (Fig. 2).

3.        Please provide the cellular effects of PAM for different time points (Ex. 4h and 24 h). For example, the authors examined signaling pathways at 24 hours, what about 4 hours?

4.        Whether PAM directly affects the effect of NAC and IWP-2.

5.        The manuscript needs to be checked for spelling (Ex. activatied? in Title), grammar, and description.

Minor suggestion:

l   Lines 20-22, “PAM” not “LTP”!?

l   Abbreviations used for the first time must provide the full name.

l   It is not the first time that microfluidic cell migration experiment has been used (Ref. 42). It is recommended to remove Fig. 1.

l   Several figures need to improve (Ex. Fig. 3. Label too small to see clearly. Fig. 4A was unclear. …).

l   Fig. 3B and Fig. 7A are redundant?

 The manuscript needs to be checked for spelling (Ex. activatied? in Title), grammar, and description.

Author Response

Reviewer: 3

  1. They used an inhibitor, IWP-2, to show PAM treatment inhibited PC-9 cell migration through the Wnt/β-catenin signaling pathway. It needs to check whether IWP-2 could really rescue the PAM-inhibited Wnt/β-catenin signaling pathway. In addition, this result needs to be validated using other lung cancer cell lines and methods such as siRNA or shRNA.

Reply: We greatly appreciated the reviewer’s detailed comments.In this experiment, we used inhibitors of Wnt/β-catenin signaling pathway, which proved that PAM inhibited the migration of PC-9 cells through Wnt/β-catenin signaling pathway. According to previous studies, it was observed that PAM could inhibit the migration and invasion of melanoma cells through Transwell and wound healing experiments. By detecting the ROS level inside and outside the cell, it was found that the ROS content inside and outside the cell increased in a dose-dependent manner with the extension of treatment time. However, after adding catalase to reduce the H2O2 level in ROS, the inhibition of migration and invasion of melanoma cells caused by PAM incubation was reversed. The Wnt/β-catenin signaling pathway was further studied, and the Wnt, β-catenin and its degradation products GSK-3β and Axin protein were detected by WB experiment. The results showed that the expression of Wnt and β-catenin was down-regulated, while the expression of its degradation products, especially GSK-3β, increased gradually with the extension of PAM treatment time. In addition, the author also intuitively showed that β-catenin fluorescence gradually weakened with the extension of PAM treatment time by immunofluorescence. It is concluded that PAM incubation can down-regulate the migration and invasion of melanoma B16 cells by regulating Wnt/β-catenin signaling pathway [1].

In addition, some studies have shown that artemisinin and its derivatives can inhibit the metastasis of NSCLC tumors by down-regulating Wnt/β-catenin signaling pathway. In this study, the author used Wnt/β-catenin signal pathway inhibitor IWP-2, and proved that the tumor inhibitory effect of artemisinin and its derivatives depended on the inactivation of Wnt/β-catenin signal pathway to some extent through WB experiment and the counting of A549 cells in G1 phase in different experimental treatment groups [2].

Based on previous studies, we designed our own experimental process. It is believed that PAM inhibits the migration of NSCLC cells to some extent by down-regulating Wnt/β-catenin signaling pathway.

Finally, thank you very much for your suggestions on this article. We deeply agree with your opinions on adding other lung cancer cell lines and further confirming the relationship between PAM and Wnt/β-catenin signaling pathway by using methods such as siRNA or shRNA. Your suggestions make the evidence chain of our experimental conclusions more rigorous. However, due to the existing conditions in our laboratory and the limited personal time and energy, it is impossible to carry out these experiments in time. In the future, we will certainly further explore the influence of PAM on other lung cancer cell lines and carry out experimental methods such as siRNA or shRNA as soon as possible in order to get a more rigorous evidence chain.

  • Liu, J.; Gao, L.; Wu, Y.; Xu, G.; Ma, Y.; Hao, Y.; … Zhang, G. Low‐temperature plasma‐activated medium inhibited invasion and metastasis of melanoma cells via suppressing the Wnt/β‐catenin pathway. Plasma Processes and Polymers. 2019.
  • Tong, Y.; Liu, Y.; Zheng, H.; Zheng, L.; Liu, W.; Wu, J.; Ou, R.; Zhang, G.; Li, F.; Hu, M.; Liu, Z.; Lu, L. Artemisinin and its derivatives can significantly inhibit lung tumorigenesis and tumor metastasis through Wnt/β-catenin signaling. Oncotarget, 2016, 7(21), 31413–31428.

  1. The explanation for why only PC-9 cells were used for most experiments. Please provide possible reasons for the different sensitivity of different lung cancer cell lines to PAM (Fig. 2).

Reply: Thank you very much for your detailed suggestions. As a kind of NSCLC cancer cell line, PC-9 cell line is more stable, more significant and more representative than other cell lines in the initial cell viability experiment and cell scratch experiment. In addition, our time and energy are limited, and it is impossible to carry out every experiment with every cell line. Therefore, we selected PC-9 cells with relatively stable experimental results, higher significant differences and more representative results.

Page 17 Line 568-Page 17 Line 580: In addition, different types of lung cancer cell lines have different sensitivities to PAM. Some scholars believe that the sensitivity of different cell lines to plasma is mainly related to the type of cell culture medium, and different components in the culture medium interact with LTP to play a corresponding role in killing tumor cells [1]. There are three hypotheses to explain the sensitivity of cancer cells to LTP treatment. It is generally believed that the killing effect of LTP on cells is through ROS, and ROS will contact the cell membrane first, so the research on the sensitivity of LTP mainly focuses on the cell membrane. The first view is that the content of aquaporin on the cell surface is the key to the sensitivity of cells to plasma [2]. Aquaporins allow ROS to enter cells and cause cell damage, so the higher the content of aquaporins on the cell surface, the stronger the sensitivity to LTP. The second view is that redox-related enzymes on cell membrane can amplify ROS produced by LTP, so the content of redox-related enzymes on cell membrane is an important reason why cells are sensitive to LTP [3]. The last point is that low cholesterol level will increase oxidation and cavity formation, and cells show higher sensitivity to LTP [4].

  • Li, Y.; Tang, T.; Lee, H.; Song, K. Cold Atmospheric Pressure Plasma-Activated Medium Induces Selective Cell Death in Human Hepatocellular Carcinoma Cells Independently of Singlet Oxygen, Hydrogen Peroxide, Nitric Oxide and Nitrite/Nitrate. Int J Mol Sci, 2021;22(11):5548.
  • Bekeschus, S.; Liebelt, G.; Menz, J.; Berner, J.; Sagwal, S. K.; Wende, K.; Weltmann, K. D.; Boeckmann, L.; von Woedtke, T.; Metelmann, H. R.; Emmert, S.; Schmidt, A. Tumor cell metabolism correlates with resistance to gas plasma treatment: The evaluation of three dogmas. Free radical biology & medicine, 2021, 167, 12–28.
  • Bauer G. Tumor cell-protective catalase as a novel target for rational therapeutic approaches based on specific intercellular ROS signaling. Anticancer Res, 2012, 32(7):2599-2624.
  • Van der Paal, J.; Hong, S. H.; Yusupov, M.; Gaur, N.; Oh, J. S.; Short, R. D.; Szili, E. J.; Bogaerts, A. How membrane lipids influence plasma delivery of reactive oxygen species into cells and subsequent DNA damage: an experimental and computational study. Physical chemistry chemical physics : PCCP, 2019, 21(35), 19327–19341.

  1. Please provide the cellular effects of PAM for different time points (Ex. 4h and 24 h). For example, the authors examined signaling pathways at 24 hours, what about 4 hours?

Reply: We greatly appreciated the reviewer’s detailed comments. First of all, we designed the experiment on the basis of previous studies. In the previous research, we found that after treating different types of cancer cells with PAM or LTP, most of them were incubated for 24 h, and then their proteins were extracted [1,2]. Therefore, in our experiment, we also use the protein after incubation for 24 h. Secondly, we also analyzed the phenomenon of 4 h cell migration in the scratch experiment, but the effect was not obvious, so it was not included in the article. In addition, the cycle of our periodic experiment is 24 h because the cell division cycle is 24 h. We think that the active substances produced by PAM treatment have just entered the cell at the time node of incubation for 4 h, and the cell has not had time to make a stress response to the active substances entering its interior, and the change of intracellular protein expression is also very weak. After incubation for 24 h, the cell has divided for a cycle and fully responded to the active substances entering the cell. The intracellular protein expression has also changed obviously, and it continues to the next generation of cells. Therefore, we think that the changes of intracellular protein expression will be more obvious and representative after PAM incubation for 24 h, which can better reflect the influence of PAM treatment on lung cancer cells.

  • Wang, Y.; Mang, X.; Li, X.; Cai, Z.; Tan, F. Cold atmospheric plasma induces apoptosis in human colon and lung cancer cells through modulating mitochondrial pathway. Frontiers in cell and developmental biology, 2022,10, 915785.
  • Liu, J.; Gao, L.; Wu, Y.; Xu, G.; Ma, Y.; Hao, Y.; … Zhang, G. Low‐temperature plasma‐activated medium inhibited invasion and metastasis of melanoma cells via suppressing the Wnt/β‐catenin pathway. Plasma Processes and Polymers. 2019. 
  1. Whether PAM directly affects the effect of NAC and IWP-2.

Reply: Thanks for reviewer’s kindly comments. According to your suggestion, in order to explore whether PAM directly affects the effect of NAC, we supplemented this by performing NAC validation experiments, and the results are shown in the following Fig. 1. Compared with the experimental group incubated with PAM, most of the extracellular ROS produced by PAM treatment were neutralized by NAC in the experimental group pretreated with NAC, and the level of extracellular ROS did not increase in a time-dependent manner with the extension of PAM treatment. It is proved that PAM treated by LTP contains ROS, and PAM directly affects the effect of NAC.

  In addition, as shown in Fig. 2 below, compared with the experimental group incubated with PAM alone, in the experimental group pretreated with Wnt/β-catenin pathway inhibitor IWP-2 before PAM incubation, the result of PAM inhibiting PC-9 cell migration was reversed. This further shows that the active substances in PAM inhibit cell migration through Wnt/β-catenin pathway in cells. In addition, the migration inhibition of cells in the experimental group treated with PAM+IWP-2 was reversed, but the migration inhibition of cells in the IWP-2 treatment group alone did not occur. To sum up, PAM did not directly affect the effect of IWP-2.

Page 14-15 Line 465-475: We also treated PC-9 cells with an inhibitor (IWP-2) of the Wnt/β-catenin signaling pathway to explore if PAM-induced inhibition of NSCLC cell migration was mediated by inhibition of Wnt/β-catenin signaling pathway. As shown in Fig. 12, a significant decrease in cell migration was observed in the 30 s PAM-treated group in Transwell experiment when compared to the control group. Compared with the 30 s PAM treatment group, the inhibition of cell migration in the IWP-2+PAM treatment group was reversed to some extent, and the number of migrated cell was significantly increased. In addition, there was no significant difference in the number of cell migration between IWP-2 and PAM+IWP-2 groups. The results demonstrated that long-term PAM treatment inhibited the migration of NSCLC cells through inhibiting the Wnt/β-catenin signaling pathway.

Fig. 1

Fig. 2

  1. The manuscript needs to be checked for spelling (Ex. activatied? in Title), grammar, and description.

Reply: We greatly appreciated the reviewer’s detailed comments. We have carefully revised the article as follows:

Page 1 Line 2: Low temperature plasma-activated medium inhibits the migration of NSCLC cells via Wnt/β-catenin pathway

Page 2 Line 61: One was to directly treat cells with LTP, and the other was that LTP first treated cell culture medium, and then incubated cells with the plasma-activated medium (PAM) [21].

  1. Lines 20-22, “PAM” not “LTP”!?

Reply: We greatly appreciated the reviewer’s detailed comments. We have carefully compared the article and revised the inappropriate expressions. The specific amendments are as follows:

Page 1 Line 21-27: The results showed that after long-term treatment with PAM, the high level of ROS produced by PAM reduced the level of mitochondrial membrane potential of cells and blocked the cell division cycle in G2/M phase. These results suggested that high ROS levels generated by PAM treatment reversed the EMT process by inhibiting the WNT/β-catenin pathway in NSCLC cells and thus inhibited the migration of NSCLC cells. Therefore, these results will provide a good theoretical support for the clinical treatment of NSCLC with PAM.

  1. Abbreviations used for the first time must provide the full name.

Reply: We greatly appreciate the reviewer’s comment. As required, we have added the full names of abbreviations that first appeared in the article, and revised them as follows:

Page 1 Line 14: This study explored the molecular mechanism of plasma activation medium (PAM) inhibiting the migration ability of NSCLC (Non-small cell lung cancer) cells.

  1. It is not the first time that microfluidic cell migration experiment has been used (Ref. 42). It is recommended to remove Fig. 1.

Reply: We greatly appreciate the reviewer’s comment. According to your suggestion, we have modified Fig.1 as follows:

Page 4 Line 167-168: Six independently controlled cell migration test units were configured on a single chip, allowing parallel testing for different conditions (Fig. 1 A and B and C).

Page 5 Line 189-194: Fig. 1. Schematic diagram of microfluidic chip structure. (A) Physical diagram of microfluidic chip. (B) Schematic diagram of 6-channel chip. (C) Single channel loading unit.①Chemokine injection port; ②Cell culture solution injection port; ③ Waste liquid outlet; ④ Cell injection port; ⑤Chemokine transport channel; ⑥Cell culture medium delivery channel; ⑦Cell transmission channel; ⑧ Waste liquid channel; ⑨ Cell isolation zone.

Fig. 1

  1. Several figures need to improve (Ex. Fig. 3. Label too small to see clearly. Fig. 4A was unclear. …).

Reply: We greatly appreciated the reviewer’s detailed comments. According to your suggestion, we have revised Fig. 3 and Fig. 4 to make the picture clearer. The revised picture is as follows:

  1. 3B and Fig. 7A are redundant?

Reply: We greatly appreciate the reviewer’s comment. According to your suggestion, we have made a more detailed and accurate description of Fig. 3B and Fig. 7A, with specific modifications as follows:

Page 8 Line 304-309: As shown in Fig. 3A-D, after continuous observation for 48 h under the light microscope, the scratch areas of cells in the short-term PAM (10 s) treatment group were significantly reduced, while those in the long-term PAM treatment group (30 s) showed no significant difference as compared with those in the control group.

Page 9 Line 329-339: Fig. 3. Effect of PAM incubation on the migration ability of NSCLC cells. (A) The wound healing experiment was conducted to detect the effect of PAM incubation on the migration of H460 cells. (B) Quantification of scratch area of H460 cells after PAM treatment. (C) Wound healing experiments were conducted to detect the effects of PC-9 cells incubated with PAM on their migration. (D) Quantification of scratch area of PC-9 cells after PAM treatment. (E) Transwell cells were used to analyze the effect of H460 cells incubated with PAM on their migration capacity. (F) The number of H460 cells passing through Transwell chamber after PAM incubation. (G) Transwell cells were used to detect the effect of PAM incubation on PC-9 cell migration. (H) The number of PC-9 cells passing through Transwell chamber after PAM incubation. Data represent the mean ± SD of three independent experiments. “ns” means no statistical difference. * p < 0.05, ** p < 0.01,*** p < 0.001 with ANOVA and 2way-ANOVA compared with the control.

Page 10 Line 362-369: To further verify that the inhibition of migration of NSCLC cells by PAM was due to the increase in ROS production caused by LTP treatment, a further set of ROS verification experiments was performed using PC-9 cells. Compared with the control group, there was no statistical difference in the area of cell scratches (Fig. 7A and 7B) and the number of cells passing through Transwell in each group (Fig. 7C and 7D) pretreated with NAC in Fig. 7. The inhibition of cell migration caused by PAM was eliminated with the addition of NAC. The evidence suggested that the migration inhibition of NSCLC cells by PAM was due to the increase of ROS induced by LTP.

Page 12 Line 390-397: Fig. 7. PAM Inhibited PC-9 Cell Migration by ROS. (A) After NAC pretreated PC-9 cells, PAM incubated the cells for 0, 12 and 48 h, and scratched the migration image. (B) Quantification of Scratch Migration in PC-9 Cells after Treatment with PAM and Incubation for 0 h, 12 h and 48 h, respectively. (C) Transwell image of PC-9 cells before PAM treatment with or without NAC pretreatment. (D) Quantification of Transwell results in PC-9 cells with and without NAC pretreatment before PAM treatment. Data represent the mean ± SD of three independent experiments. “ns” means no statistical difference. *** p < 0.001 with ANOVA and 2way-ANOVA compared with the control.

In addition, according to your suggestion, we changed some grammar problems in the article. Changes are marked in blue, as follows:

Despite the rapid development of medicine today, the threat of cancer to human health continued to increase. According to research papers published in the International Journal of Cancer, there was an estimated 18.1 million new cancer cases and 9.6 million cancer deaths worldwide in 2018 [1]. It had been reported that lung cancer was the cancer with the highest incidence and mortality in China, with the number of new cases and deaths increasing from 698,000 and 554,000 in 2015 to 823,000 and 723,000 in 2020, respectively [2]. The high mortality rate of lung cancer was closely associated with its early metastatic characteristics, which was the ability of cancer cells to leave the primary focus and infiltrated other parts of the body [3, 4]. Lung cancer can be divided into small cell lung cancer (SCLC) and non-small cell lung cancer (NSCLC), of which NSCLC accounts for about 85% of the total number of lung cancers [5]. The traditional treatments of NSCLC mainly included surgical resection, radiotherapy [6], chemotherapy [7] and targeted therapy [8]. Radiotherapy was an effective treatment for NSCLC and could be used in all stages of curative or palliative treatment [9]. Currently, platinum chemotherapy drugs is the most important chemotherapy means for advanced NSCLC patients and has a significant effect on prolonging the survival time of patients [10]. However, the inhibition effect of traditional treatment on cancer cell migration was not as expected. The existence of drug resistance and unbearable side effects forced us to find a new and effective method to inhibit NSCLC cell metastasis [11].

The plasma was that fourth state of existence of a substance consisted mainly of positively charged ion, electrons and neutral particles [12]. According to the difference of temperature, the plasma was divided into high temperature plasma and low temperature plasma (LTP) [13]. LTP had been widely used in the biomedical field in recent years due to its advantages such as low temperature and no harm to the normal tissues of the human body. It was mainly applied to such areas as wound healing [14], oral treatment [15], microbial inactivation and cancer treatment [16, 17]. At present, low-temperature plasma medicine had been widely used in dozens of cancers, such as skin cancer [18], Melanoma [19] and colon cancer [20]. There are two main ways for LTP to treat cells. One is to directly treat cells with LTP, and the other is that the culture medium was first treated by LTP, and then cells were incubated by the plasma-activated medium (PAM) [21]. Studies confirmed that the active substances in PAM could enter the cells to affect the biological effects of them [22], such as destroying cell DNA and promoting cell apoptosis [23]. The active substances in PAM mainly included reactive oxygen species (ROS) and reactive nitrogen species (RNS). Hydrogen peroxide (H2O2) and nitrite (NO2-) were the main long-acting substances of RONS, and their intracellular interaction was an important reason for the selective apoptosis of tumor cells [24]. Recently, PAM has been widely studied for anti-tumor research, proving that its production of a large number of ROS is the main factors on cancer cell apoptosis [25-27]. In addition, PAM has been found to inhibit the migration of melanoma cells [28], but whether it inhibits the migration of NSCLC and its mechanism has not been explored.

Cancer metastasis is a complex process. When cancer cells lose polarity, intercellular adhesion decreased and tight junctions are further lost, thus obtaining mesenchymal phenotype. This process was called Epithelial-Mesenchymal Transition (EMT), which was characterized by changes in levels of three prominent biomarkers (E-Cadherin, Vimentin, and N-Cadherin). E-Cadherin expression was down-regulated, while up-regulation of Vimentin and N-Cadherin means that the occurrence of EMT [29, 30]. The EMT process of cells was regulated by a variety of signaling pathways, among which the Wnt/β-catenin signaling pathway was the most closely related [31]. Besides, the Wnt/β-catenin pathway also played an important role in regulating cell proliferation, as well as cell migration and invasion [32-34]. Multiple studies had shown that the proliferation and metastasis of cancer cells such as lung cancer and bladder cancer were effectively inhibited by inhibiting the Wnt/β-catenin signaling pathway [35-39]. Furthermore, Liu et al. also proved that chromatin remodeling ATPase SMARCAD1 promoted the EMT process by activating the Wnt/β-catenin signaling pathway, thereby promoting the metastasis of pancreatic cancer cells [40]. However, whether PAM can inhibit the migration of NSCLC cells by down-regulating the Wnt/β-catenin pathway in cancer cells and inhibiting the EMT process has not been explored, and our research is dedicated to this.

In this study, the effects of PAM on the cell migration of NSCLC were detected at the in vitro level using microfluidic chips and Transwell cells. The expression levels of EMT and Wnt/β-catenin signaling pathway-related proteins were detected by Western Blot experiment to explore the internal mechanism of PAM affecting cell migration.

During the discussion, we added the following references:

  1. Bekeschus, S.; Liebelt, G.; Menz, J.; Berner, J.; Sagwal, S. K.; Wende, K.; Weltmann, K. D.; Boeckmann, L.; von Woedtke, T.; Metelmann, H. R.; Emmert, S.; Schmidt, A. Tumor cell metabolism correlates with resistance to gas plasma treatment: The evaluation of three dogmas. Free radical biology & medicine2021, 167, 12–28.
  2. Bauer G. Tumor cell-protective catalase as a novel target for rational therapeutic approaches based on specific intercellular ROS signaling. Anticancer Res, 2012, 32(7):2599-2624.
  3. Van der Paal, J.; Hong, S. H.; Yusupov, M.; Gaur, N.; Oh, J. S.; Short, R. D.; Szili, E. J.; Bogaerts, A. How membrane lipids influence plasma delivery of reactive oxygen species into cells and subsequent DNA damage: an experimental and computational study. Physical chemistry chemical physics : PCCP, 2019, 21(35), 19327–19341.
  4. Sies, H.; Jones, D.P. Reactive oxygen species (ROS) as pleiotropic physiological signalling agents. Nat Rev Mol Cell Biol, 2020, 21(7):363-383.
  5. Wang, Y.; Mang, X.; Li, X.; Cai, Z.; Tan, F. Cold atmospheric plasma induces apoptosis in human colon and lung cancer cells through modulating mitochondrial pathway. Frontiers in cell and developmental biology2022, 10, 915785.
  6. Shi, X. M.; Xu, G. M.; Zhang, G. J.; Liu, J. R.; Wu, Y. M.; Gao, L. G.; Yang, Y.; Chang, Z. S.; Yao, C. W. Low-temperature Plasma Promotes Fibroblast Proliferation in Wound Healing by ROS-activated NF-κB Signaling Pathway. Current medical science, 2018, 38(1), 107–114.
  7. Arjunan, K. P.; Clyne, A. M. Non-thermal dielectric barrier discharge plasma induces angiogenesis through reactive oxygen species. Annual International Conference of the IEEE Engineering in Medicine and Biology Society. Annual International Conference,2011, 2447–2450.
  8. Wu, H.; Zhang, Y.; Zhou, Y.; Yan, Z.; Chen, J.; Lu, T.; Song, W. Low-Dose Non-Thermal Atmospheric Plasma Promotes the Proliferation and Migration of Human Normal Skin Cells. Applied Sciences, 2023, 13(5):2866.

Round 2

Reviewer 2 Report

Authors have provided answers to all my questions and comments. I have no further issue regarding scientific aspects of this manuscript. I must admit, that revised version is very interesting and properly explain the subject, highlighting massive work that was done by authors.

If I may, I would only suggest to carefully check the manuscript as paragraph number in line 479 (discussion) and 611 (conclusion) is “4.”

Reviewer 3 Report

It's good now.